# Thermal stability and coalescence dynamics of exsolved metal nanoparticles at charged perovskite surfaces

Moritz L. Weber [1,2,3,4,16] ✉, Dylan Jennings [4,5,6], Sarah Fearn [2], Andrea Cavallaro [2,7], Michal Prochazka [1,8], Alexander Gutsche [3], Lisa Heymann [3], Jia Guo [2], Liam Yasin [2,9], Samuel J. Cooper [9], Joachim Mayer [5,10], Wolfgang Rheinheimer [4,6], Regina Dittmann [3], Rainer Waser [3,11], Olivier Guillon [4,12,13], Christian Lenser [4], Stephen J. Skinner [2], Ainara Aguadero [2,14], Slavomír Nemšák [1,15] ✉ & Felix Gunkel [3] ✉

Exsolution reactions enable the synthesis of oxide-supported metal nanoparticles, which are desirable as catalysts in green energy conversion technologies. It is crucial to precisely tailor the nanoparticle characteristics to optimize the catalysts' functionality, and to maintain the catalytic performance under operation conditions. We use chemical (co)-doping to modify the defect chemistry of exsolution-active perovskite oxides and examine its influence on the mass transfer kinetics of Ni dopants towards the oxide surface and on the subsequent coalescence behavior of the exsolved nanoparticles during a continuous thermal reduction treatment. Nanoparticles that exsolve at the surface of the acceptor-type fast-oxygen-ion-conductor $SrTi_{0.95}Ni_{0.05}O_{3-\delta}$ (STNi) show a high surface mobility leading to a very low thermal stability compared to nanoparticles that exsolve at the surface of donor-type $SrTi_{0.9}Nb_{0.05}Ni_{0.05}O_{3-\delta}$ (STNNi). Our analysis indicates that the low thermal stability of exsolved nanoparticles at the acceptor-doped perovskite surface is linked to a high oxygen vacancy concentration at the nanoparticle-oxide interface. For catalysts that require fast oxygen exchange kinetics, exsolution synthesis routes in dry hydrogen conditions may hence lead to accelerated degradation, while humid reaction conditions may mitigate this failure mechanism.

A broad implementation of a sustainable energy system requires the development of a well-aligned mix of technologies that allow for the carbon-neutral conversion, storage, and transport of energy, where chemical energy carriers such as green hydrogen are likely to play a key role across different energy sectors[1]. Active and durable catalysts are required for the electrosynthesis of green hydrogen using excess renewable energy, and for the conversion of hydrogen into electricity using electrolyzers and fuel cells. The application of nanostructured catalysts allows to considerably increase the efficiency of such devices[2,3]. Metal nanoparticles on oxide supports are among the most relevant catalysts where a high dispersion of nanoparticles results in a high density of catalytically active triple-phase boundaries.

It is particularly important to preserve the nanoparticle properties over long operation times in order to maintain the catalysts'

performance. However, limited thermal stability, caused by the coalescence of metal nanoparticles at the surface of the oxide support, often results in catalyst degradation at elevated temperatures[4–6] and hence under the operation conditions of high-temperature fuel cells and -electrolyzers or catalytic membrane reactors. The coalescence of nanoparticles is driven by the difference in the chemical potential of the nanoparticles relative to the bulk metal phase, that is related to the lower average coordination of metal atoms in nanoparticles of decreasing size[7,8].

Metal exsolution enables the synthesis of oxide-supported metal nanoparticles in a simple thermal reduction treatment of a doped parent oxide (or by using other external stimuli[9,10]). Upon thermal reduction, a fraction of the dopants is released from the oxide host lattice to the surface and nucleates in the form of metal nanoparticles. In addition to its functional properties for the composite catalyst, such as oxygen ion conductivity or electronic conductivity, the oxide serves a dual function of (1) hosting reducible cations in the parent oxide's bulk and (2) as a support for the subsequent nucleation and growth of metal nanoparticles on the surface during metal exsolution reactions. The characteristic properties of the exsolved nanoparticle population, such as the nanoparticle density as well as spatial distribution and size distribution, are a result of the mass transfer dynamics of the reducible metal species (bulk to the surface) and subsequent surface processes (nucleation, growth, and coalescence).

However, the interplay of simultaneous bulk and surface processes involved in the nanoparticle self-assembly renders one-step metal exsolution reactions rather complex. The exsolution behavior is entangled with the defect structure of the parent oxide as it determines the exsolution energetics[11–13], electrostatic interactions[14], and oxygen exchange kinetics[15]. Moreover, extended defects may serve as fast diffusion pathways[16,17] or preferential nucleation sites for exsolved nanoparticles[18,19]. Importantly, metal exsolution reactions are associated with dynamical changes in defect concentrations and their distribution within the perovskite oxide. For instance, thermal reduction applied to induce metal exsolution in perovskite oxides may result in the incorporation of vast amounts of defects (e.g. oxygen vacancies) at the perovskite surface, which serves as a support for the nanoparticles. Here, the predominant surface defect type and concentration at a given temperature and oxygen partial pressure strongly depend on the doping properties of the respective oxide. The formation of surface defects is particularly influenced by the presence of defect chemistry-dependent space-charge regions (SCRs)[20–24]. Essentially, the surface properties of donor-type STO are primarily determined by cation defects (strontium vacancies) that result in a negative surface potential over a large range of oxygen partial pressures and temperatures[14], while the surface defect structure in acceptor-type STO is primarily determined by the presence of anion defects (oxygen vacancies) that result in a positive surface potential. Therefore, surface defects are likely to impact the properties of the nanoparticle-support interface in exsolution-active perovskites of different defect chemistry. Yet, the influence of the defect structure of the perovskite oxide parent material on the nanoparticle growth and coalescence behavior of exsolved nanoparticles has not been closely examined.

This study investigates the impact of the perovskites' defect chemistry on the nanoparticle growth and coalescence behavior during metal exsolution. For this purpose, the exsolution response of Ni is compared with respect to the surface chemical and structural evolution of an acceptor-doped parent oxide and a parent oxide of effective donor-doping upon thermal reduction. Using $^{18}O$ isotopic labeling via oxygen exchange experiments, we identify oxygen vacancies to be the primary defects in STNi, while the presence of oxygen vacancies is suppressed in STNNi due to its donor-type defect chemistry. Fast Ni exsolution dynamics are detected by in situ ambient-pressure X-ray photoelectron spectroscopy for the fast-oxygen-ion-conductor STNi and correlated to the accumulation of positively charged oxygen vacancy surface defects. Most importantly, our investigations reveal pronounced differences in the coalescence behavior of the exsolved surface nanoparticles already at low temperatures of $T = 400\,°C$. Here, the presence of large concentrations of oxygen vacancies at the surface of acceptor-type perovskites during thermal reduction results in a destabilization of the (non-noble) nickel nanoparticles on the surface. In consequence, acceptor-type STNi exsolution catalysts suffer from fast particle coalescence and degradation, while additional donor-doping improves the nanoparticle stability. Our findings indicate that defect interactions at the nanoparticle-support interface considerably impact the properties of the exsolved nanoparticle population, originating from drastic differences in the predominant coalescence behavior. We demonstrate that decreasing the oxygen vacancy concentration at the surface of exsolution-active perovskites, either through tailoring the materials' defect chemistry or by control of the reaction conditions, leads to the stabilization of finely dispersed exsolved nanoparticles.

## Results

### Oxide epitaxy for high-precision studies of nanoparticle exsolution

Epitaxial thin films of $SrTi_{0.9}Nb_{0.05}Ni_{0.05}O_{3-\delta}$ (STNNi) and $SrTi_{0.95}Ni_{0.05}O_{3-\delta}$ (STNi) are employed as model systems with atomically defined perovskite surfaces to investigate nanoparticle exsolution throughout this work. Details on the epitaxial growth and the material characteristics can be found in Supplementary Note 1 of the Supplementary Information (RHEED intensity evolution, atomic force microscopy (AFM) surface analysis, high-resolution X-ray diffraction (HR-XRD) and high-resolution scanning transmission electron microscopy (HR-STEM)/energy-dispersive X-ray spectroscopy (EDXS)), as well as in the Methods section. Since the surface orientation of the perovskite parent was shown to have a major influence on the properties of the exsolved nanoparticles[25–28], both materials are synthesized in (001) orientation. In this way the metal exsolution behavior and the thermal stability of the exsolved nanoparticles can be compared at structurally identical surface facets of STNNi and STNi. A major advantage of this approach is that the influence of orientation-specific, anisotropic bulk, and surface diffusion on the exsolution behavior of the two materials can be excluded. In addition, the epitaxy of well-defined perovskite surfaces helps to minimize ambiguities regarding the influence of surface phases that are challenging to detect at the surface of porous oxides or thin films with rough surface morphology. The STNNi and STNi thin films exhibit a comparable mixed SrO / TiO₂ surface termination, as can be seen on the basis of the intensity ratio of the (0 0) specular spot and the (1 0), (−1 0) diffraction signals of the RHEED surface diffraction pattern recorded from the as-deposited samples (cf. Fig. S3)[29,30]. Therefore, a significant influence of the surface termination on the nanoparticle growth and coalescence behavior[14] can be excluded. Moreover, grain boundaries that may modify the exsolution behavior[31–33], e.g., by providing fast diffusion pathways of reducible cations in ceramic oxides as well as porosity that was reported to influence the coalescence behavior of supported nanoparticles[6,34,35] are negligible in our epitaxial catalyst model systems. The exsolution response and nanoparticle self-assembly, hence can be investigated on a single-grain level. Importantly, small inhomogeneities in the distribution of Ni dopants are detected within the as-synthesized perovskite oxide thin films by HR-STEM / EDXS (cf. Figs. S1e, S2e). This is consistent with our previous work on STNNi demonstrating partial Ni enrichment at designated areas of the perovskite lattice in the form of coherently embedded nanostructures that are accommodated within the perovskite on the basis of a domain matching mechanism[11,36]. As can be seen in Fig. S2e, similar inhomogeneities can be observed for STNi (single Ni-doped SrTiO₃), which emphasizes the importance of extended defect structures, that may serve as reservoirs of reducible transition metals in perovskite oxides.

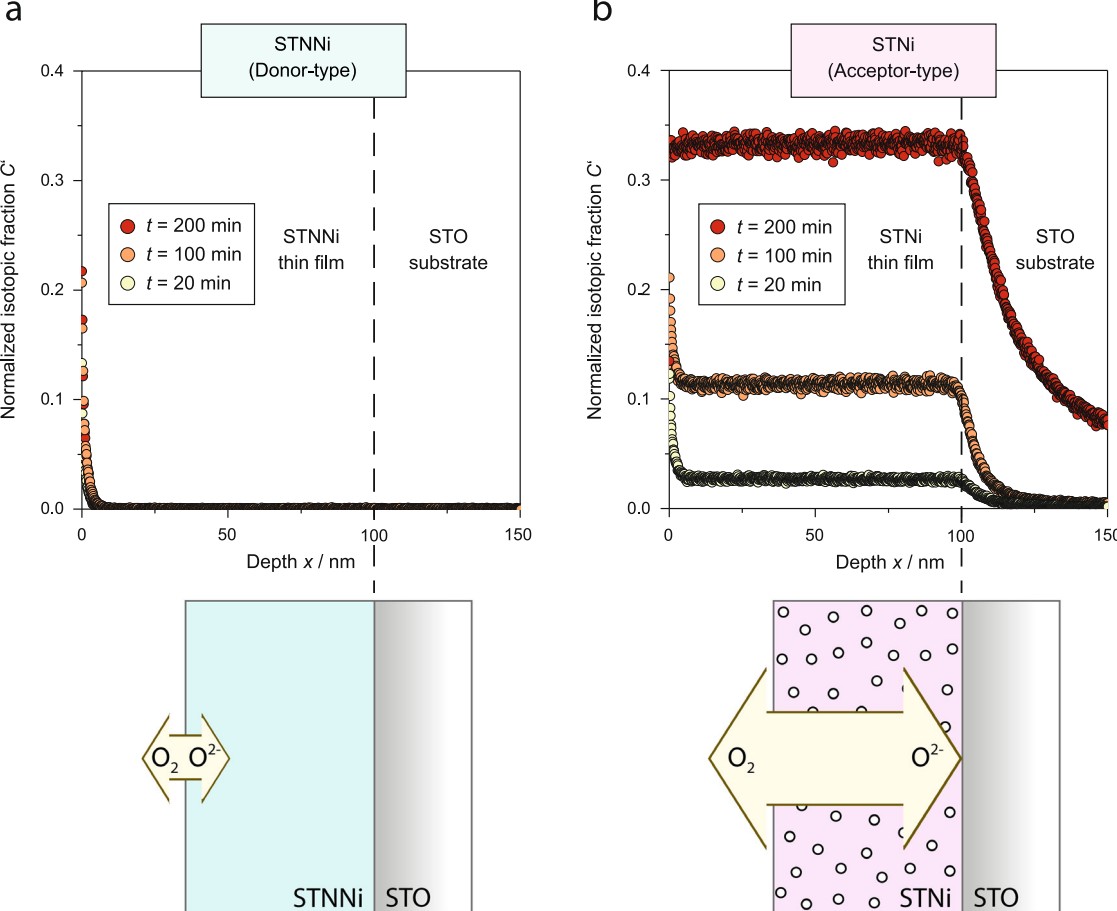

**Fig. 1 | Oxygen tracer exchange studies. a** $^{18}$O profiles determined by SIMS analysis for STNNi (donor-type) and **b** STNi (acceptor-type) after joint equilibration in $^{16}$O$_2$ gas and subsequent exchange in $^{18}$O$_2$ gas for different annealing times of $t = 20$ min, $t = 100$ min, and $t = 200$ min at $T = 400$°C and $p(O_2) = 200$ mbar. The tracer fraction $C'(x, t)$ incorporated into the thin film samples during oxygen exchange is plotted versus the sputter depth, respectively. A schematic illustration of the STNNi and STNi thin film samples is shown below respectively. STNNi exhibits a low oxygen vacancy concentration and slow correlated oxygen exchange kinetics, while STNi exhibits a large oxygen vacancy concentration and fast correlated oxygen exchange kinetics as denoted by the size and depth of the two arrows, respectively. Oxygen vacancies are depicted using circles.

It is worth noting that local inhomogeneities in the dopant distribution are likely to occur in heavily doped perovskite oxides to a certain degree. Such dopant clusters or minor phase-separated inclusions may also facilitate the formation of metal nanoparticles within the oxide bulk[11,37,38] by lowering the nucleation barrier.

## Acceptor-type and donor-type defect chemistry of STNi and STNNi

Before the metal exsolution behavior of STNNi and STNi is investigated and compared in detail, the defect chemistry of the two materials is characterized on the basis of oxygen exchange measurements. Here, the relative mobility of $^{18}$O-tracers, directly correlated to the defect chemistry of the oxides, is compared. For the $^{18}$O-tracer exchange experiments, 100 nm thick epitaxial STNNi and STNi thin film samples are jointly exchanged in $^{18}$O$_2$-enriched oxygen gas ($T = 400$ °C, $p(O_2) = 200$ mbar) with different exchange times. Subsequently, the tracer diffusion depth profiles are measured by secondary ion mass spectrometry (SIMS).

The tracer profiles reveal striking differences in the oxygen diffusion properties of STNNi and STNi. As can be seen in Fig. 1a, only small amounts of the $^{18}$O tracer are incorporated into the STNNi oxide within the investigated time-temperature window. An increased tracer fraction is detected only in the surface region (approx. 5 nm), while a steep drop in the tracer profiles is visible for larger sputter depths. In contrast, a large tracer fraction is detected in the STNi bulk (Fig. 1b). This observation is consistent with the fact that oxygen diffusion in perovskites is based on ion hopping between vacant lattice sites. The diffusion coefficient for oxygen is proportional to the oxygen vacancy concentration, $D_O \sim [V_O^{\cdot\cdot}]$, which in SrTiO$_3$ is related to the effective acceptor-concentration $2[V_O^{\cdot\cdot}] = 2[A''] + [A']$ (in Kröger–Vink notation[39]). Consequently, $^{18}$O tracers will rapidly diffuse into the thin film, resulting in a fast equilibration of the tracer concentration within the thin film bulk. While both perovskite parent oxides are doped with equal amounts of Ni acceptors (5 at.% with respect to the B-site), STNi is purely doped with Ni, therefore, oxygen vacancies are expected to counter-balance the charge of Ni dopants in acceptor-type STNi to maintain charge neutrality[40,41]. In contrast STNNi is co-doped with Nb-donors and lower-valent Ni acceptors in STNNi may be expected to be mostly counter-balanced by Nb-donors. Correspondingly, a large oxygen vacancy concentration in STNi enables fast-oxygen exchange kinetics. The STNi bulk consequently is strongly enriched by $^{18}$O isotopes, even after short exchange times and increasing levels of $^{18}$O tracers are detected with increasing annealing times. Hence, fast-oxygen exchange kinetics results in a near-uniform tracer fraction, as visible by a plateau of the detected signal between the surface region and the thin film-to-substrate interface.

The tracer profiles are modulated by the formation of space-charge regions at the perovskite surface, resulting in local deviations of

the oxygen diffusion coefficients in STNNi and STNi, which are directly coupled to the respective space-charge potential[24,42,43]. While the accumulation of oxygen vacancies is energetically favorable at the topmost surface of STNi, the extended surface SCR is strongly depleted from oxygen vacancies relative to the perovskite bulk of uniform oxygen vacancy concentration. Therefore, the perovskite is expected to exhibit a locally decreased oxygen diffusion coefficient $D_O$ in the near-surface region, that changes across the width of the extended SCR[24,42,43]. The decreased oxygen mobility in the near-surface region limits oxygen diffusion into the STNi bulk, while, after passing the confined SCR, fast bulk diffusion results in a near-flat tracer profile. The tracer profile in STNi hence is determined by the equilibrium between oxygen diffusion into the SCR and passing through the extended SCR into the bulk. Notably, a decreasing tracer profile can be observed for the diffusion of $^{18}O$ isotopes into the underlying $SrTiO_3$ substrate (Fig. 1b). In the nominally undoped substrate, the expected oxygen vacancy concentration (governed by impurity acceptor-doping[40,44,45]) is several orders of magnitude smaller, resulting in a decreased oxygen ion mobility and yielding steep tail-like diffusion profiles.

Importantly, the local enrichment of Ni acceptors at designated areas of the perovskite lattice is expected to affect the defect chemistry and related material properties of both oxides to a certain extent. Here, it is likely that the effective oxygen vacancy concentration in STNi (correlated to the acceptor-doping level) is decreased. In STNNi, partial phase separation of Ni dopants results in a surplus of Nb-donors within the perovskite host lattice. Therefore, STNNi exhibits a predominant donor-type defect chemistry[14], where the presence of oxygen vacancies is widely suppressed[20,22,23] resulting in a negligible $^{18}O$ concentration. Consistent with the $^{18}O$ tracer exchange experiments, demonstrating a direct effect of donor- and acceptor doping on the oxygen vacancy concentration, four-point probe resistivity measurements reveal finite electronic conductivity of $\sigma = 2.3$ S/cm for STNNi, while STNi is electrically insulating. The tracer exchange study demonstrates the donor-type and acceptor-type defect chemistry of STNNi and STNi, which manifests in a considerably different redox behavior and oxygen ion mobility as a consequence of the coupled surface potential and bulk defect concentrations. Here, chemical doping determines both the defect concentration profiles as well as the oxygen-exchange dynamics in the investigated STNNi and STNi thin films.

## Surface mobility and coalescence of exsolved nanoparticles

Based on the synthesis of STNNi and STNi with donor-type and acceptor-type defect chemistry and experimental verification thereof, we now turn to the investigation of the metal exsolution behavior of the two materials. The surface morphology evolution of STNNi and STNi upon thermal reduction at $T = 400\,°C$, i.e., during metal exsolution, is investigated by AFM imaging. For this purpose, several pieces were cut from the same STNNi and STNi thin film samples and jointly reduced using different annealing times ($T = 400\,°C$, $p(4\%\,H_2/Ar) = 1$ bar). A one-to-one comparison of the surface morphology of the two materials is shown in Fig. 2a, where clear differences in the exsolution behavior can be readily observed (cf. Fig. S4 for extended sets of AFM images). After short reduction times of only $t = 1$ h, the formation of an early-stage nanostructured surface is visible for STNi, while no significant nanoparticle exsolution can be observed for STNNi. After thermal reduction for $t = 2$ h, nanoparticle exsolution is also visible at the STNNi surface, whereas on equal timescales, considerable coalescence of the exsolved nanoparticles is apparent at the STNi surface. The different thermal stability of the exsolved nanoparticles at the acceptor-type and donor-type perovskite surface results in pronounced differences in the surface morphological evolution on longer timescales of $t = 17.5$ h. Here, a stable nanoparticle population of homogeneous distribution remains visible at the STNNi surface, whereas solely large agglomerates are detected at the STNi surface after equal annealing times at $T = 400\,°C$.

AFM images with smaller scan size and larger magnification obtained from STNNi and STNi after thermal reduction of $t = 2$ h are shown in Fig. 2b (equal sample state to the central panel of Fig. 2a). As can be seen, the large extended features detected at the STNi surface are formed by nanoparticle clusters. Therefore, nanoparticle coalescence at the acceptor-type STNi perovskite surface appears to originate, on the length scale accessible by AFM, predominantly from nanoparticle clustering rather than Ostwald ripening[4,46], i.e., the dissolution of smaller particles and growth of larger particles based on ionic surface diffusion. In contrast, no considerable agglomeration of nanoparticles is observed on the STNNi surface at the same time scale, indicating that the clustering of exsolved nanoparticles is much less pronounced compared to STNi and a significantly larger mobility of exsolved nanoparticles under reducing conditions at $T = 400\,°C$ at the STNi surface than in the case of STNNi. It is worth noting that we have observed Ostwald ripening to be the predominant coalescence mechanism for nanoparticles that exsolve at the STNNi surface in our previous work[14].

The different coalescence behavior is furthermore visible in Fig. 2c, which shows the evolution of the average nanoparticle and/or nanoparticle cluster diameter on a double logarithmic scale. As can be seen, the thermal instability of the nanoparticles that exsolve at the STNi surface leads to a fast increase in the average nanoparticle (cluster) size. On the same time scale, nanoparticle exsolution in STNNi results in a steady, but much less pronounced increase in the average nanoparticle size. The formation of nanoparticle clusters, however, is absent at the STNNi surface and the increase of nanoparticle size over time can be rationalized by the continuous mass transport of Ni from the subsurface to the surface[14].

Interestingly, epitaxial engineering of the surface defect chemistry of acceptor-type STNi towards donor-type doping results in a stabilization of exsolved nanoparticles at the surface (Fig. 2c and cf. Fig. S5 for AFM scans). Here nanoparticle exsolution is driven in a stack sample that consists of an STNi bottom layer (20 nm) and a Ni-free Nb:STO top layer (four monolayers), synthesized by sequential deposition using two different ceramic targets. In this way, only the surface defect chemistry of the exsolution-active acceptor-type STNi thin film is modified to exhibit donor-type character. During the reduction process, Ni dopants exsolve from the STNi bottom layer through the initially Ni-free top layer and nucleate at the surface of the stack sample. Notably, a higher temperature of $T = 600\,°C$ is required to exsolve the initial nanoparticle population at the surface, presumably to provide sufficient kinetics for the diffusion of Ni cations through the Ni-free top layer. As can be seen in Fig. 2c, no significant coalescence is detected for Ni nanoparticles that exsolved at the surface of the stack sample with acceptor-type bulk and donor-type surface characteristics. Hence, our findings show that the thermal stability of the exsolved nanoparticles at the perovskite surface is influenced by the surface defect chemistry of the respective oxide parent material. Here, the type and effective concentration of surface defects is $p(O_2)$-dependent and entangled with the perovskites defect chemistry.

These observations show that not only the mass transport step during metal exsolution towards the surface is influenced by the defect chemistry of the perovskite as we have shown in our previous publication[14], but likely also the thermal stability of the exsolved nanoparticles. Here, nanoparticles that exsolve at the perovskite surface may experience rapid coalescence even during short reduction times. In particular, we identify considerable surface mobility of Ni nanoparticles at the acceptor-type STNi surface to be the origin for particle clustering. As illustrated in Fig. 2d, the clustering process at the acceptor-type STNi surface goes along with a drop in nanoparticle density and, therefore, with a degradation of the catalytically active centers. In contrast, thermal reduction results in the exsolution of a thermally stable, homogeneously dispersed nanoparticle population at donor-type perovskite surfaces (STNNi and surface-modified STNi).

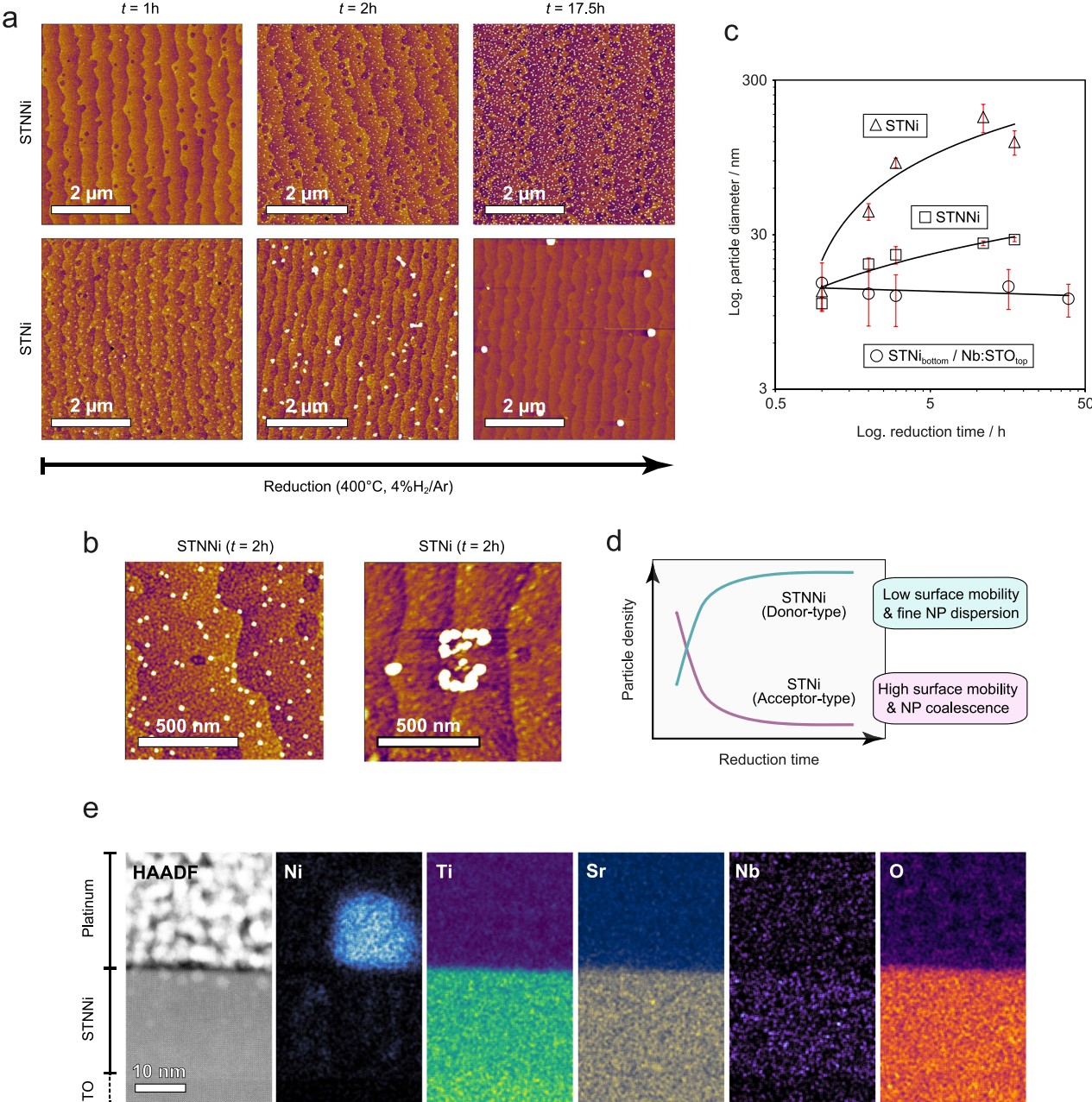

**Fig. 2 | Morphological evolution of 20 nm thick STNi and STNNi thin films after joint reduction ($T = 400\,°C$, $p(4\%\ H_2/Ar) = 1\ bar$) after different annealing times and imaging of the nanoparticle-oxide interface for reduced STNNi.**
**a** Representative surface morphology of STNNi and STNi after thermal reduction for $t = 1\ h$, $t = 2\ h$, and $t = 17.5\ h$. The scan size is $5 \times 5\ \mu m^2$. **b** Detailed scans ($1 \times 1\ \mu m^2$) obtained from STNNi and STNi after joint reduction for $t = 2\ h$ equal to the central panels shown in (**a**). **c** Influence of the reduction time on the average diameter of exsolved nanoparticles and nanoparticles clusters at the STNNi and STNi surface as well as a STNi (bottom) / Nb:STO (top) sample deposited in stack geometry. While STNNi and STNi were reduced at $T = 400\,°C$, a two-step reduction process was employed for the STNi / Nb:STO stack sample. A first annealing step was performed

at $T = 600\,°C$ for $t = 5\ h$ to generate a population of exsolved Ni nanoparticles at the surface of the stack sample. Subsequently, several consecutive annealing steps were performed at $T = 400\,°C$ with a total annealing time of $t = 39\ h$. The error bars denote the standard deviation ($\pm$ s.d.) of the averaged values. **d** Schematic illustration of the nanoparticle density evolution. In the investigated time-temperature window, donor-type STNNi exhibits a continuous exsolution reaction which results in a stable nanoparticle density. For acceptor-type STNi a rapid exsolution response is accompanied by nanoparticle clustering and a fast drop in the nanoparticle density. **e** STEM / EDXS of the nanoparticle-support interface of exsolved nanoparticles at the STNNi surface after thermal reduction ($T = 400\,°C$, $t = 200\ h$, $p(4\%\ H_2/Ar) = 1\ bar$).

Exsolved nanoparticles are frequently reported to show high robustness with respect to coalescence, which is typically assigned to a partial embedment of the metal nanoparticles into the oxide support, i.e., nanoparticle socketing[28,47–50]. Our findings show, however, that exsolved nanoparticles may still suffer from significant, material-specific coalescence effects. To investigate the role of structural anchorage via nano-socketing[28] for the thermal stability of the

exsolved nanoparticles, detailed studies of the nanoparticle-support interface are carried out by HR-STEM / EDXS analysis. Here, we focus on the Ni$_{ex}$-STNNi material system to evaluate if the increased nanoparticle stability at the STNNi surface is related to the more pronounced socketing of the exsolved Ni nanoparticles (Fig. 2e).

For this purpose, lamellae of reduced STNNi thin films ($T = 400\,°C$, $t = 200\ h$, $p(4\%\ H_2/Ar) = 1\ bar$) were prepared by focused

ion beam after the deposition of a platinum protection layer. Fig. 2e shows a Z-contrast image and corresponding EDXS maps of the STNNi surface region in cross-section geometry after long-term thermal reduction of the sample providing sufficiently large nanoparticle sizes for sample preparation when employing a low reduction temperature of $T = 400\,°C$ (for AFM image see Fig. S6a). The Z-contrast image shows an overview of the Pt-covered thin film sample, where EDXS mapping reveals the presence of an exsolved Ni nanoparticle at the thin film surface. A sharp interface between the metal nanoparticle and the oxide support with no indication for nanoparticle socketing becomes apparent. Notably, the investigation of a sample after thermal reduction at a higher temperature ($T = 900°C$, $t = 5\,h$, $p(4\%\ H_2/Ar) = 1\,bar$) also did not reveal the formation of sockets at the nanoparticle-support interface (Fig. S6b). The increased thermal stability of Ni nanoparticles that exsolve at the donor-type STNNi surface relative to nanoparticles at the acceptor-type STNi surface is hence not caused by nanoparticle socketing at the oxide support in the present case. Instead, an influence of the surface defect chemistry is apparent, where surface defects appear to play a significant role in the growth and coalescence of the exsolved nanoparticles.

## Surface chemical evolution during metal exsolution

In situ ambient-pressure X-ray photoelectron spectroscopy (AP-XPS) is carried out to correlate the defect chemistry-dependent morphological evolution observed by ex-situ AFM imaging with surface chemical changes. To investigate the metal exsolution response under reducing conditions, the donor-type STNNi and acceptor-type STNi samples were mounted side by side on the same heater ensuring an identical annealing treatment (cf. Fig. S7a). Over the course of the annealing process, the Ni 3p-Ti 3s core-level region was recorded in situ for STNNi and STNi at elevated temperatures and under reactive gas environments by alternately switching the probing position back and forth between the samples. Several consecutive annealing steps were performed, starting with a mild thermal oxidation step to desorb carbon species from the oxide surface ($T_1 = 350°C$ for $t_1 = 15\,min$ and $T_2 = 400\,°C$ for $t_2 = 10\,min$, $p(O_2) = 0.1\,mbar$), followed by a first reduction step at $T = 350\,°C$ and a second reduction step at $T = 500\,°C$ at $p(H_2) = 0.5\,mbar$, respectively. Accordingly, the exsolution response is monitored below and above the temperature that was used for the morphological investigations ($T = 400\,°C$). A photon energy of $E_{hv} = 680\,eV$ is employed to ensure high surface sensitivity, i.e., a small inelastic mean free path (IMFP) $\lambda$ -1.37 nm during the dynamic measurements under reactive gas environments and at elevated temperatures. Furthermore, static measurements were carried out (room temperature, UHV) in between the dynamic annealing steps using a photon energy of $E_{hv} = 680\,eV$ and $E_{hv} = 900\,eV$ ($\lambda$ -1.72 nm) to obtain additional chemical information from larger information depths. Examination of the Ni 3p-Ti 3s core-level signature provides information about changes in the Ni surface chemistry. In the as-prepared state of the samples, the Ni 3p spectrum is in close vicinity but well separated from the Ti 3s core-level peak (cf. Fig. S7b, c), which may serve as an internal reference to follow dynamic changes in the Ni 3p signature with respect to binding energy (BE) shifts and intensity. The Ni 3p-Ti 3s region can be deconvoluted by peak fitting of three components, namely a Ti contribution, an oxide Ni contribution ($Ni_{oxide}$), and a metal Ni contribution ($Ni_{metal}$). The peak fitting model has been empirically developed on the basis of data sets obtained from different sample states (oxidized and reduced states) as described in Supplementary Note 4.

Figure 3a, b depict relative changes in the peak areas of the oxide and metal Ni components (normalized to the Ti 3s peak area, respectively), comparing changes in the nickel chemistry at different reduction temperatures. Furthermore, changes in the binding energy over the course of the thermal reduction treatment are shown in Fig. 3c (relative to the Ti 3s position), and relative changes in the total Ni

enrichment at the surface are shown in Fig. 3d, e. Here, the bright color bars denote a high surface sensitivity, and the dark color bars denote data obtained from larger information depths, respectively.

Importantly, STNNi and STNi exhibit distinct differences in the surface Ni chemistry already after removing adventitious carbon from the air-exposed samples by mild oxidation, reflecting the characteristics of the native as-synthesized perovskite surface. Due to the high surface sensitivity of our AP-XPS measurements, the analysis very well reflects chemical differences confined within the extended SCRs at the STNNi and STNi perovskite surfaces. Here, the STNi surface is more Ni-rich (relative peak area ratio Ni 3p/Ti 3s -0.28) compared to STNNi (Ni 3p / Ti 3s -0.18) (Fig. 3d, e). The apparent difference in the Ni dopant distribution at the surface of the perovskite parent oxides is presumably attributed to the different nature of the SCR that forms during the synthesis of the thin films in oxidizing conditions, where a negative surface potential at the STNNi surface results in repulsive interactions between Ni dopants of relative negative charge (a detailed discussion can be found in ref. 14). In contrast, the positive surface potential that is characteristic for the surface of acceptor-type STNi[23,24,45] may result in an attractive force and consequently in Ni enrichment.

In addition, a larger binding energy of the Ni 3p main peak relative to the Ti 3s signal is detected for STNi (cf. Fig. 3c). Note that a larger binding energy may typically point toward an increased average oxidation state, while we refrain to assign a specific oxidation state to the Ni dopants as this is not trivial based on the analysis of complex XPS core-level spectra[51,52]. Furthermore, size-dependent particle-support interactions may result in small deviations in the absolute BE detected for the Ni nanoparticles at the STNNi and STNi surface[53,54].

In a subsequent step, the Ni chemistry is studied under reducing conditions ($T = 350\,°C$, $p(H_2) = 0.5\,mbar$). Here, STNNi shows a gradual decrease in the $Ni_{oxide}$ component (Fig. 3a) accompanied by a gradual shift in the BE towards lower values (Fig. 3c). In turn, however, only a minor increase in the $Ni_{metal}$ component is visible over the reduction time, indicating negligible metal nanoparticle exsolution in the given time-temperature window. In contrast, a fast drop in the relative signal of the $Ni_{oxide}$ component and a simultaneous formation of a $Ni_{metal}$ component is apparent for STNi (Fig. 3b), which is associated with a prompt shift in the Ni 3p BE towards lower values (Fig. 3c). These observations are consistent with the morphological investigations discussed above, indicating increased exsolution kinetics for acceptor-type STNi in comparison to the donor-type STNNi perovskite.

During the second reduction step at a higher temperature of $T = 500\,°C$, the Ni surface chemistry evolution and correlated exsolution behavior again show considerable differences for STNNi and STNi. While a strong increase in the $Ni_{metal}$ component is visible for STNNi, a drop, and ultimately, a complete loss in the $Ni_{metal}$ signal is detected at the STNi surface. Considering the morphological evolution of the perovskite surface (cf. Fig. 2), it is apparent that the large $Ni_{metal}$ component at the STNNi surface is related to a dense nanoparticle population emerging during thermal reduction at $T = 500\,°C$. For STNi, however, nanoparticles that exsolved already at lower reduction temperatures suffer from coalescence in consequence of a high surface mobility, rendering the metallic Ni phase undetectable by XPS due to loss in surface coverage. Hence, the high mobility of Ni nanoparticles and associated clustering at the STNi surface results in the observation of transient metal states already during the first reduction step at lower temperatures. Therefore, only a low intensity of metallic Ni states is detected across all spectra obtained from STNi, and relatively large uncertainties need to be considered for our quantitative analysis, while a trend in the Ni surface chemistry evolution remains visible (cf. Fig. S7).

The surface chemical evolution of the two materials hence drastically differs, which is also visible by comparing the total amount of Ni

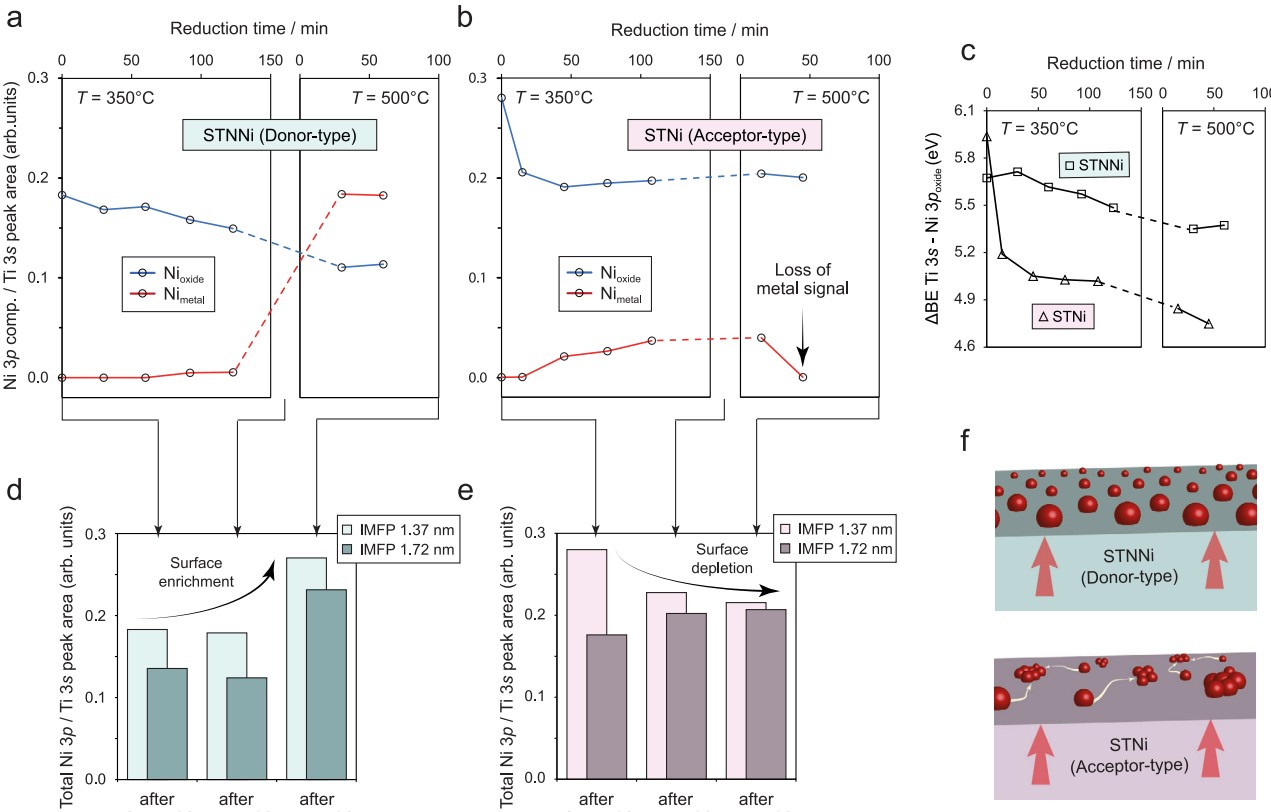

**Fig. 3 | Ambient-pressure x-ray photoelectron spectroscopy of STNNi and STNi. a, b** Temperature- and atmosphere-dependent, dynamic evolution of the Ni chemistry detected on the basis of a deconvolution model that is specified in Fig. S7 of the Supplementary Information. The analysis is based on changes in the relative fitted peak areas of the Ni 3p components normalized to the Ti 3s signal. **c** Binding energy shift of the fitted Ni components during thermal reduction. **d, e** Comparison of the total Ni 3p peak area normalized to the Ti 3s peak area after mild oxidation, low-temperature reduction at $T = 350\,°C$, and reduction at higher temperature of $T = 500\,°C$. A photon energy of $E_{hv} = 680\,eV$ is used for the analysis shown in (**a, b**), while a photon energy of $E_{hv} = 680\,eV$ and $E_{hv} = 900\,eV$ is applied for the variation of the information depth presented in (**d, e**). **f** Schematic illustration depicting the clustering behavior and correlated decrease of surface coverage of nickel nanoparticles that exsolve at the acceptor-type STNi surface relative to the exsolution of a nanoparticle population with increased thermal stability at the donor-type STNNi surface.

that is detected in the near-surface region after the consecutive thermal reduction treatments (Fig. 3d, e). Here, the Ni dopants are present either in the form of metallic nanoparticles or as oxide species (cf. Fig. S8 for respective share of the $Ni_{metal}$ and $Ni_{oxide}$ components). It is apparent that thermal reduction of the initial STNNi surface of relatively lower Ni content results in a significant enrichment of Ni at the surface and in the subsurface region (Fig. 3c). In contrast, the initially Ni-enriched STNi surface exhibits an overall decrease in the Ni content, while a minor enrichment of the subsurface region may be apparent. Notably, O 1s, Ti 2p, Nb 3d and Sr 3d core-level spectra states were periodically recorded after each of the consecutive thermal annealing steps, revealing only negligible changes in chemical environment of the Sr, Ti, and Nb cations over time (cf. Fig. S9). It is worth noting that no indications for the formation of considerable amounts of Sr-rich phases have been detected at the surface nor the bulk of the thin films upon thermal reduction (cf. Fig. S10).

Our combined morphological and surface chemical investigations (cf. Figs. 2, 3) are consistent with recent reports on the correlation of both oxygen exchange dynamics[15] and space-charge induced electrostatic interactions[14] for the metal exsolution kinetics. Here, a fast release of oxygen from the oxide parent upon thermal reduction, that can be expected for acceptor-type perovskites with fast-oxygen exchange kinetics such as STNi was reported to be correlated to a fast metal exsolution response, similar to our observations. Furthermore, we have shown in a previous study that the presence of a space-charge induced negative surface potential in donor-type perovskites

under oxidizing conditions results in a blocking character for acceptor-type cations (co-dopants) and, hence, in a delayed exsolution response. Since no such negative surface potential is present at the surface of acceptor-type STNi, a fast exsolution response in comparison to STNNi is in line with our recent report. Most striking, however, is the impact of the perovskites' defect chemistry on the coalescence behavior of exsolved nanoparticles. Our findings point towards a link between defects at the surface of exsolution-active parent materials that serve as the support for the exsolved nanoparticles and the coalescence behavior under reducing conditions. Here, considerable differences in the point defect populations at the surface of donor-type and acceptor-type STO can be expected, particularly with respect to the presence of oxygen vacancies.

## Surface-nanoparticle interactions

In order to study the general nanoparticle-support interface for differently doped oxide supports in absence of the convoluted exsolution process, metal nanoparticles are synthesized by dewetting of a sputtered Ni thin film. Dewetting results in comparably large particles with high density, which simplifies the preparation and investigation of nanoparticle cross-sections by FIB-TEM. In the following, we make use of the dewetting approach to fabricate model systems of Ni nanoparticles on single crystal substrates as a reference system for supported nanoparticles that were synthesized by metal exsolution. If the coalescence behavior is determined by surface defects, a similar trend in the coalescence dynamics for Ni nanoparticles at donor-type and

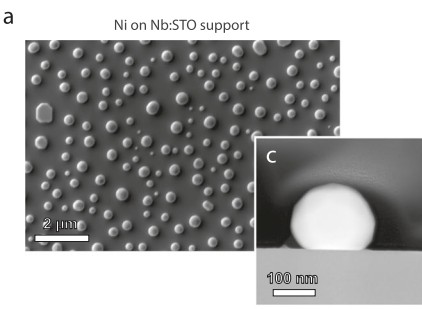

**Fig. 4 | Scanning electron microscopy and scanning transmission electron microscopy of the nanoparticle-support interface of samples that were synthesized by dewetting of sputtered 20 nm thick Ni thin films in a joint thermal reduction procedure ($T_1 = 1000\,°C$, $t_1 = 5$ h and $T_2 = 1050\,°C$, $t_2 = 23$ h in $p(4\%\,H_2/Ar) = 1$ bar). a, b** SEM imaging of Ni nanoparticles on an Nb:STO and STO substrate after dewetting. **c, d** Representative STEM cross-sections of nanoparticles on an Nb:STO and STO substrate after dewetting corresponding to the samples shown in (**a, b**). Ni particles on STO are detected to be more wetting than Ni particles on Nb:STO.

acceptor-type oxide surfaces is expected to occur during dewetting under reducing conditions as we have observed for the exsolution-active perovskites STNNi and STNi.

For this purpose, 20 nm thick Ni thin films were sputtered on TiO$_2$-terminated commercial single crystal (001) Nb:STO and (001) STO substrates (cf. Fig. S11 for as-prepared samples). SrTiO$_3$, although nominally undoped, is typically acceptor-doped by naturally occurring impurities[40,44,45]. Hence, STO can be used as a reference material to study the wetting behavior of Ni nanoparticles at acceptor-doped oxide support, while the Nb:STO substrate serves as a reference for the donor-doped oxide support. The two substrates are likely to exhibit similar types of surface defects under reducing conditions as the exsolution-active STNNi and STNi perovskites (while the concentration and distribution profiles will deviate). After dewetting of the films by high-temperature annealing (first at $T_1 = 1000°C$, $t_1 = 5$ h, $p_1(4\%\,H_2/Ar) = 1$ bar and subsequently at $T_2 = 1050\,°C$, $t_2 = 23$ h, $p_2(4\%\,H_2/Ar) = 1$ bar) scanning electron microscopy (SEM) imaging of the dewetted thin films at the Nb:STO and STO surface reveals distinct differences in the nanoparticle properties (Fig. 4a, b). Consistent with our exsolution experiments, nanoparticles fabricated at the donor-doped Nb:STO surface are more homogeneous in dispersion and in size (cf. Fig. S12 for information on the frequency distribution of the particle/cluster diameter). In contrast, nanoparticles that are formed by dewetting at the acceptor impurity-doped STO surface appear with irregular shape and size pointing towards the influence of coalescence effects.

Cross-sections of several particles were prepared from the sample, where representative STEM cross-section images are shown in Fig. 4c, d. In order to compare the relative interfacial energy of the dewetted Ni particles, a quasi-Winterbottom analysis was performed[55]. For this purpose, only single grains were considered for the analysis, where two different particles were investigated for each sample. Representative cross-section images of the nanoparticles prepared by dewetting are shown in Fig. 4c, d. As can be readily seen, the Ni particles exhibit a different wetting behavior on the differently doped substrates (Fig. 4c, d). Here, less surface wetting is visible for Ni particles on the Nb:STO substrate relative to Ni particles on the STO substrate. The difference in the wetting behavior can be further quantified based on the relative distance from the center of the Wulff shape of the nanoparticle to the particle-substrate interface, where $R_1$ is the distance from the particle-substrate interface to the Wulff center, and $R_2$ is the distance from the Wulff center to the uppermost facet of the particle (cf. Fig. S13). The ratio was determined to be $R_1/R_2$ (Ni-Nb:STO) = $0.74 \pm 0.01$ and $R_1/R_2$ (Ni-STO) = $0.50 \pm 0.04$. Here, a lower $R_1/R_2$ ratio reflects increased surface wetting, which corresponds to a lower relative interfacial energy, and consequently, a larger adhesion energy. Here, a lower interfacial energy is expected to result in an

increased energy barrier for the surface diffusion of nanoparticles at the perovskite surface, and hence in a smaller average nanoparticle size and higher nanoparticle density[27,56]. Surprisingly, however, a more pronounced surface wetting, detected for the Ni-STO material system, is correlated to a greater degree of coalescence of the supported nanoparticles under reducing annealing conditions.

It is worth noting that indications for the early formation of base structures or ridges that may ultimately evolve into nanoparticle sockets[57] were detected for the particles that have been synthesized by dewetting of a Ni thin film on STO. We provide a detailed discussion about different topological surface features that evolve under thermal reduction for the Ni-support systems in Supplementary Note 5 (cf. Fig. S14).

In all, neither the difference in adhesion energy, which is determined to be larger for Ni on the (impurity acceptor-doped) STO substrate in comparison to Ni particles on the donor-doped Nb:STO substrate, nor the formation of nanoparticle sockets can explain the different coalescence behavior. These results suggest that different processes take effect under the conditions of the thermal reduction treatment.

### Achieving control over nanoparticle stability

In order to tailor the catalytic performance of oxide-supported metal nanoparticles a precise control of the nanoparticle density, spatial distribution and size distribution is key. Our findings indicate that the nanoparticle growth and coalescence behavior, and therefore the surface characteristics of exsolution catalysts, is heavily influenced by the intrinsic and extrinsic defect structure of the oxide parent material. Previous reports have shown the critical role of oxygen vacancies and the dynamic changes in the oxygen vacancy concentration during thermal annealing under reactive gas atmospheres for the nucleation of exsolution nanoparticles[12,19] and the exsolution dynamics[14,15]. The present study demonstrates that interactions of oxygen vacancy defects at the surface of perovskite oxides (i.e., at the nanoparticle-support interface) play a critical role in the coalescence of exsolved surface nanoparticles and dynamic changes in their correlated properties upon thermal reduction. The oxygen vacancy concentration at the perovskite surface is considerably increased under reducing annealing conditions and hence during metal exsolution reactions as compared to the nominal bulk concentrations[24,45,58]. Here, the effective oxygen vacancy concentration at the perovskite surface is directly coupled to the defect chemistry of the oxide parent via the inherent nature of the respective surface SCR[24,45,58].

Since no partial embedment of the exsolved nanoparticles is observed for the present material system, there is no indication that an enhanced robustness against coalescence of the Ni particles at the donor-type STNNi surface is linked to structural anchorage. Nano-

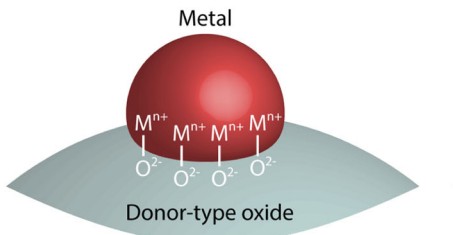

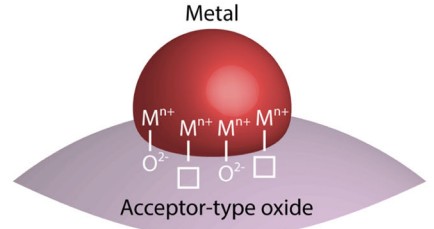

**Fig. 5 | Schematic illustration of a metal nanoparticle on an oxide support.** The stability of the particle on the surface is determined by the bond between the metal atoms of the particle and the oxygen ions of the support M-O. Here, the presence of oxygen vacancies at the particle-support interface can considerably alter the effective bonding strength of nanoparticles at oxide supports.

socketing[28], hence, does not play a role for this material system in the investigated experimental parameter space.

It is worth noting that nanoparticle socketing was shown to occur for exsolved nanoparticles in a variety of doped oxides, where the socketing process was previously attributed directly to the growth mechanism of exsolution nanoparticles by cations from the host lattice, allowing for facile interdiffusion between the two phases[28] However, the reported extent of nanoparticle socketing varies strongly across the literature and despite seminal publications on the topic[28,57], a universal mechanistic explanation on the origin of socket formation and the controllability of the socket depth and related properties is lacking. Oftentimes, A-cation deficient perovskites are used as efficient parent oxides for the synthesis of exsolution catalysts potentially enhancing the extent of interdiffusion in comparison to the stoichiometric oxides investigated in the present study. The formation of nanoparticle sockets, however, was shown to occur for Ni nanoparticles prepared on stoichiometric $BaZr_{0.9}Y_{0.1}O_3$-based oxides irrespective of the (exsolution and deposition) preparation method[26], for exsolved nanoparticles in stoichiometric $SrIr_{0.005}Ti_{0.995}O_3$ perovskites[57,59], and has been even demonstrated for A-site excess $LaFeO_3$ thin films with Pd nanoparticles being prepared by dewetting experiments[60].

However, our results can be understood in terms of particle-support interactions that typically influence the properties of supported nanoparticles[61] synthesized by a variety of (deposition) techniques. Importantly, these particle-support interactions are strongly entangled with the defect structure of the oxide support, particularly with the oxygen ion sublattice[7,62–65], and are especially important for the bond strength of metal clusters and small nanoparticles (<3 nm)[7,8] at the oxide support. The bond strength between metal nanoparticles and the oxide support is expected to be predominantly determined by the bond between the metal atoms of the particle and the oxygen ions of the support M-O (Fig. 5)[56,65]. Here, the presence of oxygen vacancies at the surface of the oxide support can considerably alter the effective bonding strength, either resulting in an increase or decrease of the thermal stability of metal nanoparticles at oxide supports. The influence of oxygen vacancies on the thermal stability of supported nanoparticles is related to the oxophilicity of the respective metal[7,66]. Oxophilic metals such as Ni bind more strongly to oxygen ions, while metals of low oxophilicity (e.g., noble metals) bind less strongly to oxygen ions. In consequence, the presence of oxygen vacancies can stabilize noble metal nanoparticles at oxide supports[66,67]. The presence of large concentrations of oxygen vacancies at the oxide surface, however, may result in a weaker particle bonding of non-noble metals at the support as the average number of M-O bonds per particle decreases relative to surfaces of low oxygen vacancy concentration. In consequence, a large oxygen vacancy concentration may result in a decreased energy barrier for the surface diffusion of nanoparticles on perovskite supports, which requires the breaking of bonds at the nanoparticle-support interface.

$^{18}O$ isotope labeling reveals a fast-oxygen exchange kinetics STNi as a result of acceptor-type doping and, hence, large concentrations of oxygen vacancies. An even larger oxygen vacancy concentration is expected to be present at the surface of acceptor-type STNi during thermal reduction, which exceeds the nominally high oxygen vacancy concentration in the perovskite bulk due to the space-charge induced accumulation of oxygen vacancies at the topmost surface. Notably, the presence of a large concentration of oxygen vacancies at the STNi surface is consistent with a high affinity for surface hydroxylation as observed by AP-XPS (cf. Fig. S9). The pronounced coalescence of exsolved nanoparticles at the STNi surface hence can be rationalized by a larger difference in chemical potential due to the strongly oxygen-deficient support under reducing conditions[7,8]. Notably, the difference in bond strength was not reflected by the wetting behavior of Ni nanoparticles at donor-type and acceptor-type substrates investigated ex situ by quasi-Winterbottom analysis, while the nanoparticle morphology clearly indicated pronounced coalescence processes at the acceptor-type surface, similar to the exsolution catalysts. It is worth noting that also the charge transfer between the supported nanoparticles and the support was reported to impact their stability[68] where the presence of oxygen vacancies at the particle-support interface as well as the doping concentrations underneath may cause a decrease in the extent of charge transfer occurring across the particle-support interface[64].

Our findings, hence, are in line with an oxygen vacancy-promoted coalescence of the exsolved Ni nanoparticles at the acceptor-type STNi surface, that mediates the effective bond strength of the nanoparticle to the support. Therefore, enhanced thermal stability of supported (exsolved and deposited) nanoparticles of oxophilic metals is expected when lowering the effective oxygen vacancy concentration at the perovskite surface, as we have demonstrated by epitaxial engineering of the surface defect chemistry (cf. Fig. 2). In addition, annealing experiments have been conducted in a humid atmosphere (cf. Fig. S15, 4% $H_2/Ar + H_2O$ as opposed to a dry 4% $H_2/Ar$ gas mixture) in order to lower the oxygen vacancy concentration at the surface of STNi by decreasing the oxygen partial pressure in the reducing gas mixture[69] and by hydroxylation of oxygen vacancies at the perovskite surface ($V_O^{\cdot\cdot} + H_2O(g) + O_O^x \rightleftharpoons 2OH_O^{\cdot}$ in Kröger–Vink notation)[70,71]. The thermal reduction in humid conditions ($T = 400\,°C$, $t = 20\,h$) results in a drastic stabilization of exsolved Ni nanoparticles at the acceptor-type STNi surface (cf. Fig. S15), allowing for the synthesis of finely dispersed nanoparticles at the acceptor-type perovskite surface. The thermal stability of exsolved nanoparticles hence, can be tailored by chemical co-doping of the perovskite parent as well as based on simple compositional modifications of the reactive gas environment applied for the exsolution process. These pathways can be employed to synthesize nanostructured metal-oxide composites with improved control over the nanoparticle properties, while combining acceptor-type or donor-type oxides with metal nanoparticles.

Our study further shows that strong similarities in the nature of supported metal nanoparticles may exist regardless of the specific preparation method e.g., with respect to defect interactions and the socketing behavior). Defect interactions may be particularly important during the early stage of exsolution reactions, since the chemical potential of the atoms in the nanoparticles is strongly size-dependent, and the increased chemical potential of small nanoparticles is likely to accelerate their coalescence[7,8]. Since particle socketing, which is often observed for exsolved nanoparticles, was shown to proceed in a subsequent step to the nucleation of exsolution nanoparticles[57], nanoparticle-support interactions may significantly influence the nanoparticle self-assembly and correlated properties in the early stage of metal exsolution reactions.

Based on the results of the present study, we hypothesize that the application of exsolution catalysts may be more suitable to specific material combinations and applications than others. In particular, applications of exsolved non-noble metal nanoparticles that require a high concentration of oxygen vacancies in the oxide to enable fast-oxygen exchange kinetics, such as electrocatalysts in high-temperature fuel cells and -electrolyzers or catalytic membrane reactors, appear challenging due to the decreased nanoparticle stability. The highly reducing environment of the cathodic reaction in electrolysis would pose a particular challenge, as the nanoparticle stability may be adversely affected by the cathodic overpotential and local gas conversion effects. Whether such effects are mitigated by the high humidity present under operation conditions remains to be investigated. In contrast, applications in heterogeneous catalysis that require no oxygen ion transport in the support may be more suitable for exsolved nanoparticles, as the enhanced stability on donor-type oxides can be utilized.

In conclusion, we have studied the metal exsolution response of Ni dopants in an acceptor-type $SrTi_{0.95}Ni_{0.05}O_{3-\delta}$ (STNi) and a donor-type $SrTi_{0.9}Nb_{0.05}Ni_{0.05}O_{3-\delta}$ (STNNi) perovskite oxide. The study has identified considerable differences in the metal exsolution dynamics of Ni towards the perovskite surface and the coalescence behavior of exsolved nanoparticles along the perovskite surface. The origin of the different metal exsolution behavior is the surface defect chemistry of the respective perovskite parent oxide. Both mass transfer of reducible dopants from the bulk to the surface, as well as subsequent surface mobility and nanoparticle coalescence, are considerably influenced by the $p(O_2)$-dependent defect population and the correlated electrostatic surface potential at the perovskite surface. Here, a much higher surface mobility of exsolved nanoparticles is revealed at the surface of acceptor-doped STNi in comparison to co-doped STNNi with effective donor-type defect chemistry. We hence find that the surface concentration of oxygen vacancies is of particular importance for the thermal stability of exsolved nanoparticles, whereas nanoparticle socketing is negligible in this material system in the investigated time-temperature window. We identify a high surface oxygen vacancy concentration under exsolution reaction conditions to be the origin of the low thermal stability of (non-noble) Ni nanoparticles at acceptor-doped perovskite surfaces, as it is associated with a decreased average number of bonds at the nanoparticle-support interface. We demonstrate that the nanoparticle characteristics in exsolution catalysts can be controlled by tailoring the oxygen vacancy concentration in the near-surface region, where chemical doping or humidification of the annealing atmosphere can be employed. The results of the present study may indicate that the application of exsolution catalysts under a strongly reducing atmosphere (such as dry hydrogen) may be limited to specific material combinations. Particularly, the fabrication of metal nanoparticles at the surface of fast-oxygen ion conductors as a material class that is interesting for applications in solid oxide cells, appears to be highly challenging. The need for a low surface oxygen vacancy concentration to stabilize exsolved nanoparticles hence requires strongly donor-type perovskites with low oxygen ion conductivity and typically high electron conductivity or a humid gas composition present in the respective device. The wet hydrogen atmosphere that is often present under SOC operation, hence might result in suppressed coalescence dynamics as a result of an increased oxygen partial pressure.

## Methods

### Epitaxial growth

[001]-oriented epitaxial thin films with thicknesses between 30 monolayers (~12 nm) and 100 nm were synthesized by PLD in orientation on $TiO_2$-terminated (001) $SrTiO_3$ or $Nb:SrTiO_3$ substrates. For ablation of the polished ceramic target, an excimer laser with a wavelength of $\lambda = 248$ nm was operated with a repetition rate of $f = 5$ Hz, and a laser fluence of $F = 1.14$ J/cm$^2$. An IR-laser was applied to control the substrates' (backside) temperature, which was $T = 650$ °C. The oxygen partial pressure was $p(O_2) = 0.108$ mbar. Stoichiometric ceramic targets, that were synthesized by the Pechini method were employed as PLD targets. An Nb:STO with 4% Nb-doping was employed to fabricate the Nb:STO (4 monolayers)/STNi (20 nm) stack samples. STNNi, STNi, and Nb:STO/STNi thin films were deposited under equal conditions and were quenched to room temperature after the growth. Line profiles were extracted from RHEED patterns recorded from as-deposited STNNi and STNi samples of different thicknesses across the (00) specular spot and the (10), (−10) diffraction spots. The analysis was performed using the image processing analysis package of IGOR Pro version 6.3.7.2.

### Thermal reduction

Ex-situ thermal reduction of the samples was performed under continuous gas flow of $p(4\%$ $H_2/Ar) = 1$ bar with a flow rate of $Q = 50$ ml/min, where the samples were quenched to room temperature in $p(4\%$ $H_2/Ar) = 1$ bar atmosphere after the reduction process. Dry (non-humidified) reduction conditions were employed throughout the main paper by default. For thermal reduction tests under humid conditions (cf. Fig. S15), the 4% $H_2/Ar$ gas flow was passed through a water bath at room temperature, before entering the quartz tube of the quench furnace. Nb:STO (4 monolayers)/STNi (20 nm) stack samples were reduced in a two-step process. A first annealing step was performed at $T = 600$°C, $t = 5$ h to accelerate cation migration through the Ni-free top layer and to generate a population of exsolved Ni nanoparticles at the surface the stack sample. Subsequently, several consecutive annealing steps were performed at $T = 400$ °C with a total annealing time of $t = 39$ h in order to evaluate the thermal stability of the exsolved Ni nanoparticles at the Nb:STO/STNi surface.

### Thin film sputtering and dewetting

About 20-nm-thick Ni thin films were sputtered on $TiO_2$-terminated (001) $SrTiO_3$ or $Nb:SrTiO_3$ single crystal substrates at a pressure of $p(Ar) = 0.0066$ mbar. A deposition rate of 20 nm/min was applied, and a power of $P = 200$ W was used. The dewetting of the sputtered Ni thin films was performed in a two-step annealing procedure ($T_1 = 1000$°C, $t_1 = 5$ h and $T_2 = 1050$°C, $t_2 = 23$ h) in a continuous flow of a $p(4\%$ $H_2/Ar) = 1$ bar gas mixture. The samples were cooled down to $T = 400$°C with a cooling rate of approximately $T = 50$°C/min before quenching to room temperature.

### Surface imaging and structural investigation

The surface morphology of each thin film was investigated by atomic force microscopy (Cypher, Oxford Instruments Asylum Research Inc., Santa Barbara, USA, and Nanosurf FlexAFM, Nanosurf AG, Liestal, Switzerland) using a probing tip with a curvature of <7 nm (Silicon-SPM-Sensor, PPP-NCHR-20, Nanosensors, Neuchatel, Switzerland). The average surface roughness of the as-prepared thin films, as well as the average nanoparticle diameter after reduction, was determined using Gwyddion 2.52. The error bars denote the standard deviation

(± s.d.) of the averaged values determined from multiple measurements, where typically three different locations of the thin films were investigated, respectively. The properties of the exsolved nanoparticle population at the Nb:STO (4 monolayers)/STNi (20 nm) stack samples were investigated by representative measurements between one and three different locations of the sample surface per annealing step, respectively. A lower threshold of 1 nm was applied to analyze the nanoparticle characteristics of all samples, where images of different scan sizes between $2 \times 2\,\mu m^2$ and $15 \times 15\,\mu m^2$ were recorded to account for different nanoparticle homogeneity and density at STNNi and STNi surfaces. The crystal structure of the epitaxial thin films was investigated by high-resolution XRD (D8 Discover, Bruker, Karlsruhe, Germany) using scans in $2\Theta\text{-}\omega$ geometry in the vicinity of the (002) diffraction peak as well as reciprocal space mapping in the vicinity of the (103) diffraction peak. The XRD system was equipped with a Goebel mirror, a Cu $K_{\alpha}$ monochromator, a centric Eulerian cradle, a Lynxeye XE detector, a divergence aperture of 0.2 mm and a pinhole adapter of 2 mm diameter.

## Transmission electron microscopy

Scanning transmission electron microscopy (STEM) analysis (including HR-STEM high-angle annular dark field (HAADF) imaging and X-ray energy-dispersive spectroscopy (EDS)) was performed with a Thermo Fisher Scientific Titan G2 80-200 CREWLEY S/TEM[72]. Transmission electron microscopy (TEM) and a part of the STEM imaging of dewetted Ni particles was done with an FEI Tecnai G2 F20[73]. Samples for STEM analysis were prepared using a Thermo Fisher Scientific Helios NanoLab 460F1 FIB-SEM[74], with a final milling energy of 2 KeV. The same microscope was also used for scanning electron microscopy (SEM) imaging of dewetted Ni films. Select liftouts were further polished using a Fischione Model 1040 NanoMill, with a final polishing energy of 500 eV. Analysis of EDS data was done using Hyperspy[75]. The relative surface energy of the dewetted Ni particles was compared using a quasi-Winterbottom analysis[55]. Given that the particles are not aligned down low-index zone axes with the substrate, a true Winterbottom analysis was not possible. However, the relative distance from the center of the Wulff shape of the crystal to the Ni/substrate interface was measured, which can then be compared between the two samples (which were treated identically) to give a measure of the differences in relative interfacial energy between the two systems. The standard deviation (± s.d.) obtained for the measurement of two particles for each material system denotes the error for the determined $R_1/R_2$ values.

## Electrical characterization

Four-point probe resistivity measurements were performed at room temperature using an AC/DC Hall Effect Measurement System Model 8404 (Lake Shore Cryotronics Inc., Ohio, USA). For this purpose, thin films are deposited on $SrTiO_3$ substrates and contacted to 10 mm solder pad sample cards by ultrasonic Al-wire bonding (Kulicke & Soffa Industries Inc., Singapore).

## Isotopic labeling and SIMS depth profiling

In order to jointly equilibrate the thin film samples prior to the tracer exchange experiments, 100-nm-thick STNNi and STNi thin film samples were placed in a quartz tube. The tube was evacuated to $p < 10^{-7}$ mbar before it was filled with $^{16}O_2$ gas ($p = 200$ mbar) and heated to $T = 400\,°C$. After cooling down to room temperature, the gas environment was exchanged with $^{18}O_2$ enriched oxygen gas, without exposing the samples to air, and a thermal annealing step was carried out for different exchange times of $t = 20$ min, $t = 100$ min, and $t = 200$ min. The equilibration step was performed for $t_{\text{equilibration}} \geq 10 \times t_{\text{exchange}}$, respectively. The samples were transferred directly into the hot zone of the furnace to ensure fast heating rates and were transferred to the cool zone (room temperature) of the furnace for quenching down the sample temperature after the exchange experiment. SIMS depth profiling was carried out with positive polarity and by using burst mode[76]. Depth-profiles of $^{16}O^-$, $^{18}O^-$ (and $Ni^-$, $NiO^-$, $NbO^-$, $SrO^-$, $TiO^-$) were extracted from the mass spectra. The depth profile measurements were performed on an IONTOF ToF-SIMS V instrument. The primary ion beam was a 25 keV $Bi^+$ beam at 0.2 pA, and the analytical region was $100 \times 100\,\mu m^2$ collecting negative secondary ions. The depth profiling was carried out with a 1 keV $Cs^+$ beam over a $300 \times 300\,\mu m^2$ area. A flood gun was used for charge compensation. The measured tracer fraction in the thin films $C(x, t)$ was corrected by the background concentration (natural abundance $C_{\text{bg}} = 0.2\%$ and the isotope enrichment $C_{\text{g}} = 83.45\%$ of the gas used for the exchange experiments, to calculate the tracer fraction $C'(x, t)$ incorporated into the samples during oxygen exchange. The percentage of $^{18}O$ isotope enrichment of the $^{18}O_2$-rich gas was determined by annealing a silicon wafer at $T = 1000\,°C$ for $t = 8$ h and subsequent measurement of the $^{16}O/^{18}O$ ratio by SIMS analysis. The sputter depth was calculated from the sputter time by normalization to the respective nickel dopant profiles, which are equal to the sample thickness of 100 nm as controlled by RHEED monitoring.

## In situ spectroscopy

Ambient-pressure X-ray photoelectron spectroscopy was performed at beamline 9.3.2 of the Advanced Light Source at Lawrence Berkeley National Laboratory using soft X-rays and with a background pressure of $p \sim 1 \times 10^{-10}$ mbar in the vacuum chamber. The hemispherical electron analyzer (Scienta R4000 HiPP) was differentially pumped in order to maintain ultra-high vacuum conditions at the detector. To allow for a direct comparison of STNNi and STNi during identical annealing conditions, $10 \times 10\,mm^2$ samples were cut into $5 \times 10\,mm^2$ pieces and mounted side by side on the sample heater. During the measurements, a thermocouple was connected to the STNNi sample. The thin film thickness of the two samples was 20 nm. First a cleaning step by mild thermal oxidation was performed ($T_1 = 350\,°C$ for $t_1 = 15$ min and $T_2 = 400\,°C$ for $t_2 = 10$ min, $p(O_2) = 0.1$ mbar). Subsequently, a first reduction step was performed ($T = 350\,°C$, $p(H_2) = 0.5$ mbar) and a second reduction step was performed at $T = 500\,°C$, $p(H_2) = 0.5$ mbar). The sample was quenched down to room temperature in the respective gas environment after each annealing step. Subsequently, the chamber was evacuated. The samples were characterized under static conditions before the next annealing step was initiated. The Ni 3$p$-Ti 3$s$ core-level region was recorded alternatingly for STNNi and STNi by switching the probing position between the samples after the annealing temperature stabilized. The scan time was approximately 15 min for each spectrum. In order to vary the information depth, different photon energies were employed for the analysis. A photon energy of $E_{\text{hv}} = 680$ eV was used for the analysis during reducing annealing i.e., under dynamic conditions (ambient pressure, elevated temperatures). Ni 3$p$-Ti 3$s$, O 1$s$, Ti 2$p$, C 1$s$, Nb 3$d$, Sr 3$d$ core-level spectra were periodically recorded under static conditions (UHV, room temperature) using a photon energy of $E_{\text{hv}} = 680$ eV and $E_{\text{hv}} = 900$ eV. The IMFP was calculated on the basis of the Ni 3$p$ core-level energy with the QUASES-IMFP-TPP-2M Ver. 3.0 software using the TPP-2M formula[77]. For this purpose, the electronic structure of as-prepared STNNi and STNi was considered, while the density of $SrTiO_3$ was used. Deconvolution of the XPS spectra by peak fitting was performed with KolXPD using Voigt peak shapes of fixed widths, fixed relative positions, fixed magnitude of spin-orbit-splitting, and branching ratio (for Ni 3$p$ components). A Shirley-type background was subtracted. The binding energy of the Ni 3$p$-Ti 3$s$ region was corrected with respect to BE(Ti 3$s$) = 62.0 eV. Additional information on the fitting procedure is given in the Supplementary Information (cf. Fig. S7).

## Data availability

The data supporting the findings of this study are available within the paper and its Supplementary Information file. The experimental data are available in the Jülich DATA repository at https://doi.org/10.26165/JUELICH-DATA/QZ8NOS.

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

## Acknowledgements

This research used beamline 9.3.2 of the Advanced Light Source, a US DOE Office of Science User Facility under Contract No. DE-AC02-05CH11231. M.L.W. was supported in part by an ALS Collaborative Postdoctoral Fellowship. M.L.W. acknowledges receiving funding from the German Academic Exchange Service (DAAD) within the frame of a research fellowship for doctoral students. M.L.W. and F.G. sincerely thank C. Baeumer and M. Nguyen for providing sputtered Ni thin film samples, as well as Terry McAfee, Grigory Potemkin, and Sylvia de Waal for their experimental support.

## Author contributions

M.L.W. performed the sample synthesis, sample processing as well as XRD and AFM analysis. D.J. performed the STEM/EDXS analysis of the epitaxial thin films and the sputtered thin films after dewetting. M.L.W. carried out the isotopic labeling experiments with support from A.C. S.F. supported the SIMS analysis. M.L.W. conducted the AP-XPS analysis, where M.P. and S.N. supported the experiments. A.G. and L.H. supported the AFM imaging. J.G. supported the annealing experiments. M.L.W. evaluated the experimental data with contributions, including in-depth discussions, from D.J., S.F., L.Y., S.J.C., C.L., S.J.S., A.A., S.N. and F.G. M.L.W., S.N., and F.G. determined the research direction and conceptualized the experiments. J.M., W.R., R.D., R.W., O.G., S.J.S., A.A., S.N., and F.G. supervised the research. M.L.W. conceived and wrote the original manuscript and edited the manuscript with contributions from all authors. All authors reviewed the manuscript and have given approval to the final version of the manuscript.

## Funding

## Competing interests

The authors declare no competing interests.

## Additional information

[1]Advanced Light Source, Lawrence Berkeley National Laboratory, Berkeley, CA 94720, USA. [2]Department of Materials, Imperial College London, London SW7 2AZ, United Kingdom. [3]Peter Gruenberg Institute for Electronic Materials (PGI-7) and Juelich-Aachen Research Alliance (JARA-FIT), Forschungszentrum Juelich GmbH, 52425 Juelich, Germany. [4]Institute of Energy Materials and Devices, Materials Synthesis and Processing (IMD-2), Forschungszentrum Juelich GmbH, 52425 Jülich, Germany. [5]Ernst Ruska-Centre for Microscopy and Spectroscopy with Electrons, Materials Science and Technology (ER-C 2), Forschungszentrum Juelich GmbH, 52425 Juelich, Germany. [6]Universität Stuttgart, Institute for Manufacturing Technologies of Ceramic Components and Composites (IFKB), 70569 Stuttgart, Germany. [7]Department of Engineering and Applied Sciences, University of Bergamo, 24044 Dalmine, Italy. [8]New Technologies Research Centre, University of West Bohemia, 301 00 Pilsen, Czech Republic. [9]Dyson School of Design Engineering, Imperial College London, London SW7 2DB, United Kingdom. [10]Central Facility for Electron Microscopy (GFE), RWTH Aachen University, 52064 Aachen, Germany. [11]Institute for Electronic Materials (IWE 2), RWTH Aachen University, 52074 Aachen, Germany. [12]Institute of Mineral Engineering (GHI), RWTH Aachen University, 52062 Aachen, Germany. [13]Juelich-Aachen Research Alliance (JARA-Energy), 52425 Juelich, Germany. [14]Instituto de Ciencia de Materiales de Madrid (ICMM-CSIC), 28049 Madrid, Spain. [15]Department of Physics and Astronomy, University of California, Davis, California, CA 95616, USA. [16]Present address: Next-Generation Fuel Cell Research Center, Kyushu University, 744 Motooka, Nishi-ku, Fukuoka 819-0395, Japan. ✉e-mail: weber.lukas.moritz.320@m.kyushu-u.ac.jp; SNemsak@lbl.gov; f.gunkel@fz-juelich.de

