## [Transparent Peer Review file · Nature Communications]

Thermal stability and coalescence dynamics of exsolved metal nanoparticles at charged perovskite surfaces

Corresponding Author: Dr Moritz Weber

Version 0:

Reviewer comments:

Reviewer #1

(Remarks to the Author)

In this manuscript, the authors investigated the metal exsolution response of Ni dopants in the p-type $\text{SrTi}_{0.95}\text{Ni}_{0.05}\text{O}_{3-\delta}$ (STNi) and n-type $\text{SrTi}_{0.9}\text{Nb}_{0.05}\text{Ni}_{0.05}\text{O}_{3-\delta}$ (STNNi) perovskite oxide. They discovered the exsolved nanoparticles on the surface of acceptor-doped STNi exhibit much higher surface mobility than those on the surface of STNNi with effective donor-type defect chemistry, because less surface concentration of oxygen vacancies is positive for the thermal stability of exsolved nanoparticles. Therefore, the nanoparticle characteristics in exsolution catalysts can be controlled by tailoring the oxygen vacancy concentration via chemical doping or humidification of the annealing atmosphere. This is an interesting work with solid methodologies and well-described results, which will be instructive for designing stable exsolved nanoparticles for thermocatalysis and electrocatalysis. Minor revisions are necessary before acceptance for publication:

1. Nanoparticles that exsolve at the surface of the p-type STNi show a very low thermal stability compared to nanoparticles that exsolve at the surface of n-type STNNi, which means the Ni nanoparticles are more stable on STNNi. Considering that no nanoparticle socketing at the oxide support of STNNi, what is the main reason for the improvement of stability?
2. Why is the ^{18}O tracer fraction in STNi (Figure 1) kept almost constant in the depth from 10 to 100 nm? It is different from the trend in STO, which decreases with the depth.
3. The doping of Nb descends the concentration of oxygen vacancies. Does its effect suppress the exsolution rate of Ni, the overall exsolution ratio of Ni, or improve the stability of the interface?
4. For STNi and STNNi, the exsolution of B-site Ni from the host lattice to the surface will cause the excess of Sr at A site, so what is the change of A-site Sr with the exsolution of Ni? Will it segregate on the surfaces of STNi and STNNi or form another perovskite-type structure?
5. How about the O 1s XPS spectra of STNNi and STNi during the ambient pressure x-ray photoelectron spectroscopy test in Figure 3?
6. In Figure S9, why does the peak intensity at around 532 eV in O 1s XPS spectra of STNi increase after reduction in H_2 at 500 °C?
7. While I understand, that the use of model systems is necessary to infer the conclusions as stated in the manuscript, I would appreciate if the authors could try to bridge the gap between the model systems and perovskite materials that are closer to actual use.

Reviewer #2

(Remarks to the Author)

In this work, the exsolution behavior of Ni nanoparticles (NPs) on a p-type $\text{SrTi}_{0.95}\text{Ni}_{0.05}\text{O}_{3-\delta}$ (STNi) and n-type $\text{SrTi}_{0.9}\text{Nb}_{0.05}\text{Ni}_{0.05}\text{O}_{3-\delta}$ (STNNi) smooth surfaces with an identical orientation of (001) was investigated in details. It is found that in comparison to the Ni nanoparticles exsolved from STNNi surface, the Ni NPs exsolved from STNi surface have a lower coalescence resistance (a lower thermal stability). Through various characterization techniques, it is clearly proven that the oxygen vacancy concentration at the Ni NP/STNi perovskite interface is much higher than that at the Ni NP/STNNi interface so that the bonding strength between Ni NPs and STNi perovskite is much weaker than that between Ni NPs and STNNi perovskite, resulting in a lower thermal stability. While most of the previously reported work associates the thermal

stability of exsolved NPs with NP socketing, this work points out that NP socketing does not play a role in determining the thermal stability of NPs exsolved from STNNi and STNi perovskites. This work provides a new insight into the feasibility of exsolution strategy. In particular, in the devices where a high oxygen vacancy concentration is required, applications of exsolved NPs appear challenging due to the decreased thermal stability of NPs. Considering the high quality and novelty of this work, I recommend its publication in Nature Communications after minor revision. Below are some suggestions which may be useful for future improvement.

1. If possible, could the authors explain why NP socketing phenomenon is not detected on the Ni NPs exsolved from STNi and STNNi perovskites.
2. Is it possible to use any method to measure the bonding strength between the NPs and parent perovskite?
3. On Lines 398-399, Page 19, the authors stated "Representative cross-section images of the nanoparticles prepared by dewetting are shown in Fig. 4d, e". "Fig. 4d, e" should be "Fig. 4c, d".
4. On Lines 389-390, Page 19, the authors stated "Consistent with our exsolution experiments, nanoparticles fabricated at the donor-doped Nb:STO surface are more homogeneous in dispersion and in size". Some quantitative analysis should be provided to support this statement. For example, the size distributions of NPs in "Fig. 4a, b" should be analyzed and compared.
5. Similarly, the contact angle should be marked in Fig. 4c, d to support the statement "Here, a larger contact angle and lower surface wetting is visible for Ni particles on the Nb:STO substrate relative to Ni particles on STO substrate."
6. If possible, the type (p-type or n-type) of STNi and STNNi should be experimentally confirmed.

Reviewer #3

(Remarks to the Author)

The manuscript deals with the concept of exsolution as a way of produced nanostructured electrocatalyst and particularly with the coalescence dynamics of Ni exsolved nanoparticles as function of the oxygen surface mobility. Two different epitaxially grown host perovskites are employed for this study i.e. a p-type $\text{SrTi}_{0.95}\text{Ni}_{0.05}\text{O}_{3-\delta}$ (STNi) and a n-type $\text{SrTi}_{0.9}\text{Nb}_{0.05}\text{Ni}_{0.05}\text{O}_{3-\delta}$ (STNNi) with identical (001) surface orientation to study the mass transfer kinetics of Ni dopants towards the oxide surface and in the subsequent coalescence behavior of the exsolved nanoparticles at the perovskite surface during a continuous thermal reduction treatment.

Authors have used an impressive set of advanced techniques to study the coalescence dynamics such as: (i) isotopic labeling experiments (i.e. $^{18}\text{O}_2$) combined with secondary ion mass spectrometry to study the oxygen mobility, (ii) AFM imaging during thermal reduction with dry hydrogen as function of the process duration to study the evolution of particle size and density, (iii) STEM/EDXS to study the nanoparticle support interfaces after reduction, (iv) AP-XPS to correlate the defect chemistry morphological evolution observed by AFM with the surface chemical changes and (v) surface nanoparticle interactions using dewetting of Ni sputtered thin films.

The main conclusion of the manuscript as presented in the: (i) abstract is that the low thermal stability of exsolved nanoparticles is associated to a large oxygen vacancy concentration at the nanoparticle oxide interface, hampering the applicability of the exsolution synthesis route for catalysts that require a fast oxygen exchange kinetics, (ii) conclusions is that the fabrication of metal nanoparticles (via exsolution) at the surface of fast oxygen ion conductors as a material class that is interesting for applications in solid oxide cells, appears to be highly challenging.

Overall the study suggests that exsolution is not a promising concept for the development of stable nanostructured catalysts used in realistic conditions especially in the field of solid oxide electrolysis cells. I find that this as argument is not in line with the studies of this work as well as with the literature and the rational follows below: (i) The concept of exsolution in solid oxide cells has also find application in oxygen electrodes (see <https://doi.org/10.1021/acs.nanolett.5b04160>; <https://doi.org/10.1002/adfm.202001326>) where operating conditions are oxidative and thus the introduction should clarify that the findings of this work are not applicable for this case. (ii) Exsolution can be triggered by several other ways i.e. plasma, light or ion irradiation (see <https://doi.org/10.1002/smt.202100868>; <https://doi.org/10.1039/D3EE02448B>; <https://doi.org/10.1021/acsnano.2c05128>) these studies should be part of the introduction and should be connected with the findings of this work. (iii) Finally, the biggest issue is that at dry hydrogen reduction conditions the nanoparticles of STNi (i.e. a perovskite with high oxygen mobility) tend to agglomerate as function of time, however this is not the case (i.e. nanoparticles are stable) when reduction is done at realistic conditions i.e. in presence of humidity (see Fig. S13). Authors mention in connection to Fig. S13 the following "Notably, dry reduction conditions are employed for all measurements throughout the main paper leading to fast coalescence of exsolved Ni nanoparticles. For humidification of the gas mixture the 4% H_2/Ar gas flow was passed through a water bath at room temperature before entering the quartz tube of the quench furnace. As can be seen, the exsolution behaviour changes dramatically, where pronounced coarsening is visible for dry conditions and finely dispersed nanoparticles decorate the p-type STNi surface after thermal reduction in a humidified gas atmosphere." This observation seems to cancel the main conclusions of the manuscript.

There is no doubt that this work has very interesting findings however the connection of observations with the conclusions is problematic (see Fig. S13) as well as their connection with literature (i.e. triggering of exsolution and applications). Taking into account the above my suggestion is to that this work is suitable for another journal after addressing the aforementioned points.

Reviewer #4

(Remarks to the Author)

The topic is highly relevant to the field of materials science, especially for applications in green energy conversion technologies. The focus on nanoparticle stability and dynamics at the interface of charged perovskite surfaces is timely and of potential interest for researchers working on catalysis, solid state ionics, chemical and electrochemical membranes etc.

The paper addresses a significant research gap by elucidating the impact of perovskite defect chemistry on the growth and coalescence behavior of exsolved nanoparticles. The comparative study between acceptor-doped and donor-doped perovskites offers original insights into how defect chemistry influences nanoparticle dynamics, which is less explored in current literature.

The paper is well-structured and written with clarity, effectively guiding the reader through the study's rationale, methods, results, and conclusions. The coherent presentation of complex concepts and results facilitates understanding. The figures and plots are of high quality, effectively illustrating key findings and supporting the textual analysis. They are well-integrated into the text, enhancing the paper's overall clarity and impact.

Data analysis is meticulous, with a clear correlation between defect chemistry and nanoparticle behavior elucidated through experimental results. The interpretation of data, especially the differences in mass transfer kinetics and the coalescence behavior between STNi and STNNi, is insightful and convincingly argued.

Overall, I believe this is a very high quality manuscript and a significant advance in understanding in the field of exsolved materials and nanotechnology for producing robust nanoparticles. I recommend publication, subject to minor points below.

A key aspect that I believe that authors have not made entirely clear is that particle socketing and epitaxy is intimately linked to the use of A-site deficient perovskite (see eg statements on P12). A-site stoichiometric perovskite, as used in the current study, will form other types of metal particles interfaces since B-site exsolution induces the formation of AO phases and/or Ruddlesden Popper phase, thus significantly altering the interface structure and composition and thus stability. Indeed, some of the supplementary data such as Fig S11 indicates that the particle-substrate interface obtained here is atypical of particles exsolved from deficient perovskites. This does not detract from the novelty, but provides a more nuanced discussion around the importance of defect chemistry support for achieving particle stability.

Additionally, while the paper effectively correlates defect chemistry with nanoparticle dynamics, it could benefit from a deeper mechanistic insight. Specifically, the discussion around charge carrier concentration and its impact on surface energy modifications could be expanded. Including theoretical models or simulations that support the experimental observations could enrich the discussion.

Version 1:

Reviewer comments:

Reviewer #1

(Remarks to the Author)

The authors have revised the manuscript accordingly, and I recommend its acceptance for publication.

Reviewer #2

(Remarks to the Author)

The authors have well addressed the comments raised during the first round of the reviews and I find that the revised manuscript meets the acceptance criteria and recommend its acceptance.

Reviewer #3

(Remarks to the Author)

The authors addressed in their revised version most of the points of my assessment, but there is a major topic which has not yet been resolved. The authors use the experimental findings presented in the main text of this work (and not the findings of the supporting information) to make generalized conclusions and strong statements about the limited applicability of exsolution synthesis route for electrodes that require high oxygen mobility. However, this generalized conclusion lacks foundation and is not appropriately justified.

To be more specific, my concern relates to the following phrases which appear in the abstract and conclusion sections.

Abstract: Our analysis indicates that the low thermal stability of exsolved nanoparticles at the acceptor-doped perovskite surface is associated to a large oxygen vacancy concentration at the nanoparticle-oxide interface, hampering the applicability of the exsolution synthesis route for catalysts that require a fast oxygen exchange kinetics.

Conclusions: The results of the present study may indicate that the application of exsolution catalysts may be limited to specific material combinations, where the fabrication of metal nanoparticles at the surface of fast oxygen ion conductors as a material class that is interesting for applications in solid oxide cells, appears to be highly challenging. The need for a low surface oxygen vacancy concentration to stabilize exsolved nanoparticles hence requires strongly donor-type perovskites with low oxygen ion conductivity and typically high electron conductivity or modifications in the respective fuel composition applied in the respective device.

I strongly believe that the message that the authors are trying to convey is inappropriate and misleading, for the following reasons. First of all, the present study does demonstrates the low stability of the exsolved particles on perovskites with high

oxygen mobility, but only when unrealistic conditions are applied i.e. dry hydrogen environment. The dry hydrogen conditions are unrealistic since the exsolution concept with high oxygen mobility perovskites is mainly applicable for oxygen conducting solid oxide cells (SOCs). However, the fuel electrode in SOCs typically involves wet conditions, for instance: (i) in the fuel cell mode of SOCs, water is formed in the fuel electrode during operation, (ii) in the water electrolysis mode of SOCs, hydrogen and water co-exist, (iii) in the carbon dioxide electroreduction mode of SOCs, water is in the feed. In oxygen electrodes, exsolved nanoparticle decorated electrodes operate in the absence of hydrogen and thus the findings of this work are also not applicable.

Moreover, the results of the present study in wet hydrogen conditions (presented in the supplementary information), which are the realistic scenario, showed that exsolved nanoparticles are stable in both evaluated classes of materials (i.e. with high and low oxygen mobility).

Thus, the results of the present study do not support the generalized statements made in the abstract and conclusions and thus I cannot recommend the publication of this work. Based on these points, I would invite the authors to make a second revision if they think that they can address these concerns and if not my recommendation would be the rejection of this work.

Reviewer #4

(Remarks to the Author)

The authors have addressed my comments in full. I recommend publication.

Version 2:

Reviewer comments:

Reviewer #3

(Remarks to the Author)

Authors have addressed all my concerns.

Point-by-point response letter regarding revision of manuscript NCOMMS-23-62974

We sincerely thank the referees for their efforts and expertise providing thorough reviews of our manuscript. Below, we give a point-by-point response to all remarks.

Reply to Reviewer #1:

Reviewer #1: In this manuscript, the authors investigated the metal exsolution response of Ni dopants in the p-type $\text{SrTi}_{0.95}\text{Ni}_{0.05}\text{O}_{3-\delta}$ (STNi) and n-type $\text{SrTi}_{0.9}\text{Nb}_{0.05}\text{Ni}_{0.05}\text{O}_{3-\delta}$ (STNNi) perovskite oxide. They discovered the exsolved nanoparticles on the surface of acceptor-doped STNi exhibit much higher surface mobility than those on the surface of STNNi with effective donor-type defect chemistry, because less surface concentration of oxygen vacancies is positive for the thermal stability of exsolved nanoparticles. Therefore, the nanoparticle characteristics in exsolution catalysts can be controlled by tailoring the oxygen vacancy concentration via chemical doping or humidification of the annealing atmosphere. This is an interesting work with solid methodologies and well-described results, which will be instructive for designing stable exsolved nanoparticles for chromatolysis and electrocatalysis. Minor revisions are necessary before acceptance for publication:

- We thank the reviewer for their very positive feedback.

(1.1) *Nanoparticles that exsolve at the surface of the p-type STNi show a very low thermal stability compared to nanoparticles that exsolve at the surface of n-type STNNi, which means the Ni nanoparticles are more stable on STNNi. Considering that no nanoparticle socketing at the oxide support of STNNi, what is the main reason for the improvement of stability?*

- The main reason for the larger thermal stability of exsolved Ni nanoparticles at the STNNi perovskite surface is the low surface oxygen vacancy concentration associated with Nb donor doping. Nickel nanoparticles form strong bonds with the oxygen ions of an oxide support due to the non-noble character and corresponding high oxophilicity of Ni. When a large concentration of oxygen vacancies is present, however, the average number of bonds at the nanoparticle-support interface is decreased. This is the case for acceptor-doped STNi, where the effective nanoparticle-support bonding strength is decreased, which is expected to lower the activation energy for particle migration.

We would like to kindly refer to line 450 – line 469 of our original manuscript providing a detailed discussion of the origin of the low thermal stability of non-noble Ni nanoparticles on acceptor-doped oxide surfaces. To further clarify our discussion, we have added the following statement in the conclusion section of our revised manuscript (line 559 – line 562).

“We identify a high surface oxygen vacancy concentration under exsolution reaction conditions to be the origin of the low thermal stability of (non-noble) Ni nanoparticles at acceptor-doped perovskite surfaces, as it is associated with a decreased average number of bonds at the nanoparticle-support interface.”

(1.2) *Why is the ^{18}O tracer fraction in STNi (Figure 1) kept almost constant in the depth from 10 to 100 nm? It is different from the trend in STO, which decreases with the depth.*

- The reason for the near-uniform ^{18}O tracer profile detected for STNi is the large acceptor doping concentration of the oxide. In general, the diffusion coefficient for oxygen is proportional to the oxygen vacancy concentration, $D_{\text{O}} \sim [V_{\text{O}}^{\bullet}]$, which in SrTiO_3 is related to the effective acceptor-concentration $2[V_{\text{O}}^{\bullet}] = 2[A'] + [A']$. Therefore, the intentionally acceptor-doped STNi layer exhibits a diffusion constant that is orders of magnitude larger compared to the (impurity-doped) STO substrate. Consequently, ^{18}O tracers will rapidly diffuse into the thin film resulting in a fast equilibration of the tracer concentration within the thin film bulk becoming apparent in form of a close-to uniform concentration, while a slight gradient persists as a fraction of the tracer diffuses into the underlying STO substrate. In comparison, the defect chemistry of the nominally undoped STO substrate is

determined by naturally occurring acceptor impurities of comparably low concentration. Hence, a lower oxygen vacancy concentration provides for finite oxygen ion mobility. Therefore, ^{18}O profiling reveals a strongly decreasing amount of tracers in the STO substrate with increasing sputter depth.

To clarify, we have edited the discussion in our revised manuscript (line 139 – line 143).

“The diffusion coefficient for oxygen is proportional to the oxygen vacancy concentration, $D_{\text{O}} \sim [V_{\text{O}}^{\bullet}]$, which in SrTiO_3 is related to the effective acceptor-concentration $2[V_{\text{O}}^{\bullet}] = 2[A''] + [A']$. Consequently, ^{18}O tracers will rapidly diffuse into the thin film resulting in a fast equilibration of the tracer concentration within the thin film bulk becoming apparent in form of a close-to uniform concentration.”

Furthermore, we have edited our statement in the revised manuscript to improve the clarity of our discussion (line 166 – line 168).

“In the nominally undoped substrate the expected oxygen vacancy concentration (governed by impurity acceptor-doping^{40,43,44}) is several orders of magnitude smaller, resulting in a decreased oxygen ion mobility and yielding steep tail-like diffusion profiles.”

(1.3) *The doping of Nb descends the concentration of oxygen vacancies. Does its effect suppress the exsolution rate of Ni, the overall exsolution ratio of Ni, or improve the stability of the interface?*

- We thank the reviewer for their comment. Yes, the perovskites' defect chemistry and correlated surface oxygen vacancy concentration directly impacts the exsolution behaviour with respect to the exsolution dynamics (*i.e.* exsolution rate) and the effective bonding strength (*i.e.* stability) of the nanoparticle-support interface. We would like to highlight that these are the key findings of our study.

Here, a high oxygen vacancy concentration in acceptor-doped STNi results in a fast oxygen exchange kinetics and a positive surface charge, resulting in a faster exsolution dynamics compared to donor-type STNNi. The large oxygen vacancy concentration at the surface of acceptor-doped STNi, however, results in a decreased bonding between the oxophilic Ni nanoparticles and the oxide support (see also comment 1.1). For a detailed discussion on the role of electrostatic interactions for the metal exsolution kinetics, that is governed by charged surface defects, we kindly refer to our previous publication [1].

If the overall amount of Ni exsolved at the donor-doped STNNi and acceptor-doped STNi surface (*i.e.* exsolution ratio) significantly differs, however, is difficult to answer due to the strong differences in the nanoparticle coalescence behaviour. In particular, the formation of nanoparticle clusters of complex shape at the STNi surface renders the determination of the sum of the exsolved nanoparticle volume challenging. Therefore, we refrain from making any definite statement on this matter.

(1.4) *For STNi and STNNi, the exsolution of B-site Ni from the host lattice to the surface will cause the excess of Sr at A site, so what is the change of A-site Sr with the exsolution of Ni? Will it segregate on the surfaces of STNi and STNNi or form another perovskite-type structure?*

- We thank the reviewer for their comment. The question on defect compensation mechanisms coming into play during metal exsolution reactions is highly relevant.

We present additional XPS and HR-STEM data in Response Figure R1 to demonstrate that no significant Sr-rich phases form at the thin film surface or in the thin film bulk upon thermal reduction in the investigated time-temperature window. For this purpose, we show the change in the relative peak area ratios of A-site and B-site cations before and after thermal reduction, respectively (Fig. R1a, b). It can be seen, that the change in the A / B-ratio is detected to be negligible for both STNNi (Fig. R1a) and STNi (Fig. R1b) after the thermal reduction treatment indicating that surface SrO formation as a consequence of B-site cation exsolution is negligible.

Furthermore, we show HR-STEM imaging of the thin films, comparing the as-prepared state and the thermally reduced state of representative STNNi and STNi samples (Fig. R1c-f). As can be seen, a coherent perovskite lattice is visible for all samples. We, therefore, can exclude degradation effects in form of phase segregation and the formation of extensive amounts of extended defects within the perovskite lattice, as was demonstrated in the literature e.g. for $\text{La}_{0.6}\text{Sr}_{0.4}\text{FeO}_{3-\delta}$ ^{2,3}.

Based on our analysis, we hypothesize that the exsolution of B-site Ni dopants may be predominantly compensated by the formation of Ruddlesden-Popper type defects (or to a lesser extent by the formation of B-site vacancies), as opposed to the decomposition of the perovskite structure, which limits the release of Sr cations.

Response Figure R1. (a, b) Relative peak area ratio or A-site / B-site cations obtained from STNNi and STNi thin films after mild oxidation and after thermal reduction at $T = 500^\circ\text{C}$ using a photon energy of $E_{\text{hv}} = 680 \text{ eV}$ for the analysis. (c-f) High resolution transmission electron microscopy images comparing STNNi and STNi thin films in the as-prepared state and after thermal reduction. The inset images show a magnification of $\sim 10 \times 10$ unit cells, each obtained from the main images.

We have included Response Figure R1 as Figure S10 in the Supporting Information of our revised manuscript to improve the clarity of our analysis, and we have added the following statement in our revised manuscript (line 359 – line 361).

“It is worth noting that no indications for the formation of considerable amounts of Sr-rich phases have been detected at the surface nor the bulk of the thin films upon thermal reduction (*cf.* Fig. S10).”

Accordingly, the following discussion was added to Supplementary Note 4.

“Fig. Fig. S10 shows additional XPS and HR-STEM data to demonstrate that no significant Sr-rich phases form at the thin film surface or in the thin film bulk upon thermal reduction in the investigated time-temperature window. A comparison of the relative peak area ratios of A-site and B-site cations before and after thermal reduction are shown in Fig. S10a, b respectively. As can be seen no significant changes in the respective A/B-ratio is detected for both STNNi (Fig. S10a) and STNi (Fig. S10b) after the thermal reduction treatment indicating that surface SrO formation as a consequence of B-site cation exsolution is negligible. Furthermore, HR-STEM imaging of the

thin films in the as-prepared state and the thermally reduced state of representative STNNi and STNi samples is shown in (Fig. S10c-f). As can be seen, a coherent perovskite lattice is visible for all samples. We, therefore, can exclude degradation effects in form of phase segregation and the formation of extensive amounts of Sr-rich extended defects within the perovskite lattice. Based on our analysis, we hypothesize that the exsolution of B-site Ni dopants may be predominantly compensated by the formation of Ruddlesden-Popper type defects (or to a lesser extent by the formation of B-site vacancies), as opposed to the decomposition of the perovskite structure, which limits the release of Sr cations.”

(1.5) *In Figure S9, why does the peak intensity at around 532 eV in O 1s XPS spectra of STNi increase after reduction in H₂ at 500 °C?*

- The O 1s core-level is dominated by two main peaks. The peak at lower binding energy can be assigned to the perovskite lattice, while the peak at larger binding energy is related to the presence of a variety of surface hydroxides. Due to the similar chemical environment of different hydroxide species, several components may contribute to the mixed hydroxide peaks causing a broadening as well as slight asymmetry of the mixed hydroxide peak at lower binding energy. The exact shape and intensity of the peak hence depends on the surface composition and residual hydroxylation of the surface. As shown in our manuscript, the thermal reduction of STNi at $T = 500^\circ$ will be associated with rapid nanoparticle coalescence and, hence, considerable surface chemical changes of the sample. Minor changes in the hydroxylation behaviour must be expected as a consequence of the altered surface chemistry (e.g. due to the fact that more SrO / TiO₂ planes will be exposed when the nanoparticle coverage is reduced), likely becoming apparent in form of slight changes in the O 1s signature.

(1.6) *While I understand, that the use of model systems is necessary to infer the conclusions as stated in the manuscript, I would appreciate if the authors could try to bridge the gap between the model systems and perovskite materials that are closer to actual use.*

- We thank the reviewer for their valuable comment, and we agree that the transfer of our findings to applied materials, *i.e.* ceramic oxides, is very relevant to gain a full understanding of the thermal stability limitations in exsolution catalysts.

We want to emphasize that our study provides a first fundamental understanding of the defect chemistry-dependent coalescence behavior of exsolved nanoparticles, which has been so far widely neglected in the field. We provide these first insights, on the basis of “single-grain level” analyses, revealing a drastic effect of surface oxygen vacancies on the nanoparticle stability. Based on our work we intend to point the scientific community to the significance of this issue.

As we have stated in our original manuscript (line 95 – line 112), and as has been acknowledged by the reviewer, a model system approach was necessary to enable a study solely on the influence of the defect chemistry, while excluding secondary effects that may affect the results. Comparable studies on ceramic oxides, *i.e.* in presence of grain boundaries, different crystallographic orientations, surface curvature, porosity and surface roughness hence pose a major challenge that will require dedicated investigations. While highly relevant, however, a study of ceramic oxides is beyond the scope of our already quite extensive work.

Reply to Reviewer #2:

Reviewer #2: In this work, the exsolution behavior of Ni nanoparticles (NPs) on a p-type SrTi_{0.95}Ni_{0.05}O_{3-δ} (STNi) and n-type SrTi_{0.9}Nb_{0.05}Ni_{0.05}O_{3-δ} (STNNi) smooth surfaces with an identical orientation of (001) was investigated in details. It is found that in comparison to the Ni nanoparticles exsolved from STNNi surface, the Ni NPs exsolved from STNi surface have a lower coalescence resistance (a lower thermal stability). Through various characterization techniques, it is clearly proven that the oxygen vacancy concentration at the Ni NP/ STNi perovskite interface is much higher than that at the Ni NP/STNNi interface so that the bonding strength between Ni NPs and STNi perovskite is much weaker than that between Ni NPs and STNNi perovskite, resulting in a lower thermal stability. While most of the previously reported work associates the thermal stability of exsolved NPs with NP socketing, this work points out that NP socketing does not play a role in determining the thermal stability of NPs exsolved from STNNi and STNi perovskites. This work provides a new insight into the feasibility of exsolution strategy. In particular, in the devices where a high oxygen vacancy concentration is required, applications of exsolved NPs appear challenging due to the decreased thermal stability of NPs. Considering the high quality and novelty of this work, I recommend its publication in Nature Communications after minor revision. Below are some suggestions which maybe useful for future improvement.

➤ We thank the reviewer for their very positive feedback.

(2.1) *If possible, could the authors explain why NP socketing phenomenon is not detected on the Ni NPs exsolved from STNi and STNNi perovskites.*

➤ Nanoparticle socketing was shown to occur for exsolved nanoparticles in a variety of doped oxides, where the socketing process was previously attributed directly to the growth mechanism of exsolution nanoparticles by cations from the host lattice, allowing for facile interdiffusion between the two phases⁴. However, the reported extent of nanoparticle socketing varies strongly across the literature⁵⁻⁹ and despite seminal publications on the topic^{4,8}, a universal mechanistic explanation on the origin of socket formation and the controllability of the socket depth and related properties is lacking.

(2.2) *Is it possible to use any method to measure the bonding strength between the NPs and parent perovskite?*

➤ We thank the reviewer for their helpful comment. While the bonding strength between metal nanoparticles and oxide supports is delicate to measure, it is possible to calculate the interfacial energy as an indicator for the bonding strength based on a Winterbottom analysis¹⁰ (cf. discussion in line 394 – line 413 of our original manuscript). However, our analysis shows that the difference in interfacial energies (i.e. wetting behavior) obtained by *ex-situ* experiments cannot explain differences in the nanoparticle mobility under the conditions of the reducing thermal treatment. Here, *in-situ* studies of the evolution in nanoparticle shape and interfacial properties of the particle-support system as a function of the reaction atmosphere and temperature will be required to resolve the dynamic material response at the particle-support interface under relevant reaction conditions. Our findings will help to tailor such experiments for future investigations.

(2.3) *On Lines 398-399, Page 19, the authors stated “Representative cross-section images of the nanoparticles prepared by dewetting are shown in Fig. 4d, e”. “Fig. 4d, e” should be “Fig. 4c, d”.*

➤ We sincerely thank the reviewer for pointing this out. We have revised the respective statement.

(2.4) *On Lines 389-390, Page 19, the authors stated “Consistent with our exsolution experiments, nanoparticles fabricated at the donor-doped Nb:STO surface are more homogeneous in dispersion and in size”. Some quantitative analysis should be provided to support this statement. For example, the size distributions of NPs in “Fig. 4a, b” should be analyzed and compared.*

- To corroborate our analysis, we show the size distributions for Ni nanoparticles fabricated by dewetting of sputtered Ni thin films, corresponding to Fig. 4 of our original manuscript. As can be seen, a unimodal frequency distribution is detected for Ni particles on the donor-type Nb:STO substrate, while a non-normal frequency distribution becomes apparent for Ni particles fabricated on the impurity acceptor-doped STO substrate. Here, an increased number of large Ni particles / clusters points towards pronounced coalescence effects. Notably, the increased coalescence effects during dewetting of Ni on STO, are correlated to a lower nanoparticle density of $1.2 \mu\text{m}^{-2}$ (versus $1.7 \mu\text{m}^{-2}$ for dewetting of Ni on Nb:STO).

Response Figure R2. Frequency distribution of the nanoparticle / cluster diameter of Ni nanoparticles fabricated by dewetting of a sputtered Ni thin film on a Nb:STO and a STO single crystal substrate ($T_1 = 1000^\circ\text{C}$, $t_1 = 5 \text{ h}$ and $T_2 = 1050^\circ\text{C}$, $t_2 = 23 \text{ h}$ in $p(4\% \text{ H}_2/\text{Ar}) = 1 \text{ bar}$). The particle / cluster diameters are determined based on scanning electron microscopy, using an average of three measurements for each particle, respectively.

To improve the clarity of our analysis, we included Response Figure R2 as Supplementary Figure S12 in our revised manuscript and edited the referenced statement now reading as follows (line 400 - line 403).

“Consistent with our exsolution experiments, nanoparticles fabricated at the donor-doped Nb:STO surface are more homogeneous in dispersion and in size (cf. Fig. S12 for information on the frequency distribution of the particle / cluster diameter).”

Accordingly, we have added the following discussion in Supplementary Note 5.

“The size distributions for Ni nanoparticles, fabricated by dewetting of sputtered Ni thin films, corresponding to Fig. 4 of the main manuscript is shown in Fig. S12. As can be seen, a unimodal frequency distribution is detected for Ni particles on the donor-type Nb:STO substrate, while a non-normal frequency distribution becomes apparent for Ni particles fabricated on the impurity acceptor-doped STO substrate. Here, an increased number of large Ni particles / clusters points towards pronounced coalescence effects. Notably, the increased coalescence effects during dewetting of Ni on STO, are correlated to a lower nanoparticle density of $1.2 \mu\text{m}^{-2}$ (versus $1.7 \mu\text{m}^{-2}$ for dewetting of Ni on Nb:STO).”

(2.5) Similarly, the contact angle should be marked in Fig. 4c, d to support the statement “Here, a larger contact angle and lower surface wetting is visible for Ni particles on the Nb:STO substrate relative to Ni particles on STO substrate.”

- We thank the reviewer for pointing this out and we agree that our statement needs further clarification. For a qualitative comparison of the wetting behavior of supported metal nanoparticles, we discuss R_1/R_2 values in our original manuscript. The notation refers to the Wulff shape of the

supported nanoparticles, where R_1 is the distance from the particle-substrate interface to the Wulff center, and R_2 is the distance from the Wulff center to the uppermost facet of the particle (cf. Fig. S11 of the original manuscript). The reason is that R_1/R_2 values more accurately describe differences in the surface wetting behaviour of supported metal nanoparticles, which typically show a certain extend of faceting that may considerably influence the perceived contact angle (more accurate e.g. for liquids).

Therefore, we have removed the term *contact angle* in our manuscript, while comparing difference in surface wetting exclusively based on the respective R_1/R_2 values. Accordingly, our statement now reads as follows (line 412 – line 417).

“Here, less surface wetting is visible for Ni particles on the Nb:STO substrate relative to Ni particles on the STO substrate. The difference in the wetting behavior can be further quantified based on the relative distance from the center of the Wulff shape of the nanoparticle to the particle-substrate interface, where R_1 is the distance from the particle-substrate interface to the Wulff center, and R_2 is the distance from the Wulff center to the uppermost facet of the particle (cf. Fig. S13). [...]”

(2.6) *If possible, the type (p-type or n-type) of STNi and STNNi should be experimentally confirmed.*

➤ We thank the reviewer for their helpful suggestion.

Based on the reviewers' comment, we have realized that the wording used in our manuscript may be misleading and we want to emphasize that we initially used the term “p-type” simply to describe acceptor doping and “n-type” to describe donor doping of the perovskite oxides. However, we did not intend to make a statement about the electrical properties of the STNi and STNNi perovskites (i.e. implying n-type or p-type electrical conductivity). To improve the clarity of our discussions, we have revised the manuscript with respect to all instances where “p-type / n-type” are used and replaced them with “acceptor-type / donor-type”.

The different types of doping in STNNi and STNi is clearly demonstrated on the basis of our ^{18}O tracer exchange experiments as discussed in our original manuscript. Here, a fast oxygen exchange kinetics associated to its' large oxygen vacancy concentration is detected for acceptor-type STNi, while a low oxygen exchange kinetics is detected for donor-type STNNi as a result of a low oxygen vacancy concentration.

These findings are consistent with room temperature four-point probe resistivity measurements, performed to characterize the electrical properties of STNNi. For this purpose, a 50 nm thick STNNi thin film was deposited on a 10x10 mm SrTiO_3 substrate. A sheet resistivity of $\rho_s = 80000 \text{ } \Omega/\text{sq}$ was determined, corresponding to a conductivity of about $\sigma = 2.3 \text{ S/cm}$. Hence, STNNi exhibits finite n-type electronic conductivity (room-temperature p-type hole conductivity can be typically excluded for STO based oxides¹¹). In comparison, STNi is electronically insulating, which is a direct consequence of acceptor-doping and charge compensation by oxygen vacancies¹¹.

To improve our discussion, we have added the following statement in our revised manuscript (line 175 - line 178).

“Consistent with the ^{18}O tracer exchange experiments, demonstrating a direct effect of donor- and acceptor doping on the oxygen vacancy concentration, four-point probe resistivity measurements reveal finite electronic conductivity of $\sigma = 2.3 \text{ S/cm}$ for STNNi, while STNi is electrically insulating.”

Accordingly, we have updated the *Methods* section in our revised manuscript (line 646 – 649).

“Four-point probe resistivity measurements were performed at room temperature using an AC/DC Hall Effect Measurement System Model 8404 (Lake Shore Cryotronics Inc., Ohio, USA). For this purpose, thin films are deposited on SrTiO_3 substrates and contacted to 10 mm solder pad sample cards by ultrasonic Al-wire bonding (Kulicke & Soffa Industries Inc., Singapore).”

Reply to Reviewer #3:

Reviewer #3: The manuscript deals with the concept of exsolution as a way of produced nanostructured electrocatalyst and particularly with the coalescence dynamics of Ni exsolved nanoparticles as function of the oxygen surface mobility. Two different epitaxially grown host perovskites are employed for this study i.e. a p-type $\text{SrTi}_{0.95}\text{Ni}_{0.05}\text{O}_{3-\delta}$ (STNi) and a n-type $\text{SrTi}_{0.9}\text{Nb}_{0.05}\text{Ni}_{0.05}\text{O}_{3-\delta}$ (STNNi) with identical (001) surface orientation to study the mass transfer kinetics of Ni dopants towards the oxide surface and in the subsequent coalescence behavior of the exsolved nanoparticles at the perovskite surface during a continuous thermal reduction treatment. Authors have used an impressive set of advanced techniques to study the coalescence dynamics such as: (i) isotopic labeling experiments (i.e. $^{18}\text{O}_2$) combined with secondary ion mass spectrometry to study the oxygen mobility, (ii) AFM imaging during thermal reduction with dry hydrogen as function of the process duration to study the evolution of particle size and density, (iii) STEM/EDXS to study the nanoparticle support interfaces after reduction, (iv) AP-XPS to correlate the defect chemistry morphological evolution observed by AFM with the surface chemical changes and (v) surface nanoparticle interactions using dewetting of Ni sputtered thin films. The main conclusion of the manuscript as presented in the: (i) abstract is that the low thermal stability of exsolved nanoparticles is associated to a large oxygen vacancy concentration at the nanoparticle oxide interface, hampering the applicability of the exsolution synthesis route for catalysts that require a fast oxygen exchange kinetics, (ii) conclusions is that the fabrication of metal nanoparticles (via exsolution) at the surface of fast oxygen ion conductors as a material class that is interesting for applications in solid oxide cells, appears to be highly challenging.

➤ We thank the reviewer for their very positive feedback on the quality of our work.

(3.1) *Overall the study suggests that exsolution is not a promising concept for the development of stable nanostructured catalysts used in realistic conditions especially in the field of solid oxide electrolysis cells. I find that this as argument is not in line with the studies of this work as well as with the literature and the rational follows below: (i) The concept of exsolution in solid oxide cells has also find application in oxygen electrodes (see <https://doi.org/10.1021/acs.nanolett.5b04160>; <https://doi.org/10.1002/adfm.202001326>) where operating conditions are oxidative and thus the introduction should clarify that the findings of this work are not applicable for this case.*

➤ We thank the reviewer for their comment. However, we would like to point out, that the findings presented in the literature on supported Ag nanoparticles cited by the reviewer are in full agreement with our findings. As we discuss in our manuscript, the thermal stability of metal nanoparticles on acceptor-doped supports depends on the oxophilicity of the metal. Non-noble metals, such as Ni, are highly oxophilic and form stable bonds with the oxygen ions of the oxide support. Therefore, they are more stable on a support with a low oxygen vacancy concentration. In contrast, noble metals such as Ag exhibit a low oxophilicity and hence are more stable on supports with a high oxygen vacancy concentration. The findings of our study hence indicate that the synthesis of supported nanoparticles *via* metal exsolution is more challenging for certain material combinations, while it certainly brings along various benefits compared to conventional routes such as infiltration. We kindly refer to our discussion in line 450 – line 464 of our original manuscript and the references therein for more information on the matter.

To improve the clarity of our manuscript, we have revised discussion which now reads as follows (line 534 - line 539).

“Based on the results of the present study, we hypothesize that the application of exsolution catalysts may be more suitable to specific material combinations and applications than others. In particular, applications of exsolved **non-noble metal** nanoparticles that require a high concentration of oxygen vacancies in the oxide to enable fast oxygen exchange kinetics, such as electrocatalysts in high temperature fuel cells and -electrolyzers or catalytic membrane reactors, appear challenging due to the decreased nanoparticle stability.”

(3.2) (ii) Exsolution can be triggered by several other ways i.e. plasma, light or ion irradiation (see <https://doi.org/10.1002/smt.202100868>; <https://doi.org/10.1039/D3EE02448B>; <https://doi.org/10.1021/acs.nano.2c05128>) these studies should be part of the introduction and should be connected with the findings of this work.

- We thank the reviewer for their suggestions. We have added a brief statement in the introduction of our manuscript including two of the references pointed out by the reviewer (line 36 – line 37). The application under reducing operation conditions will have a similar impact on all exsolution catalysts, mainly depending on the specific material combination, and independent from the pathway of triggering the exsolution reaction. Therefore, we expect that our results can be readily transferred to the material systems discussed in the referred literature.

“Metal exsolution enables the synthesis of oxide-supported metal nanoparticles in a simple thermal reduction treatment of a doped parent oxide (or by using other external stimuli^{12,13}).”

(3.3) (iii) Finally, the biggest issue is that at dry hydrogen reduction conditions the nanoparticles of STNi (i.e. a perovskite with high oxygen mobility) tend to agglomerate as function of time, however this is not the case (i.e. nanoparticles are stable) when reduction is done at realistic conditions i.e. in presence of humidity (see Fig. S13). Authors mention in connection to Fig. S13 the following “Notably, dry reduction conditions are employed for all measurements throughout the main paper leading to fast coalescence of exsolved Ni nanoparticles. For humidification of the gas mixture the 4% H₂/Ar gas flow was passed through a water bath at room temperature before entering the quartz tube of the quench furnace. As can be seen, the exsolution behaviour changes dramatically, where pronounced coarsening is visible for dry conditions and finely dispersed nanoparticles decorate the p-type STNi surface after thermal reduction in a humidified gas atmosphere.” This observation seems to cancel the main conclusions of the manuscript.

- We politely disagree with the reviewer’s opinion that our finding that humidity impacts the exsolution behaviour in acceptor-doped oxides “cancels the main conclusions of the manuscript”. A large variety of reaction conditions are employed throughout the literature reporting on the exsolution behaviour of oxides, where the nanoparticle density is one of the most commonly used metrics for the comparison of different materials. It is of utmost importance to understand that these metrics may be corrupted already by slight changes in the annealing conditions to allow for a reliable evaluation of the nanoparticle yield. Our study first provides a detailed explanation of the origin of the effect of humidity on the exsolution behaviour and provides a mechanistic basis for utilizing modifications in the annealing atmosphere to control the metal exsolution behaviour in oxides. In fact, we show that the correct choice of exsolution conditions can help to suppress fast coalescence and catalyst deactivation in predominantly acceptor-doped oxides.

(3.4) There is no doubt that this work has very interesting findings however the connection of observations with the conclusions is problematic (see Fig. S13) as well as their connection with literature (i.e. triggering of exsolution and applications). Taking into account the above my suggestion is to that this work is suitable for another journal after addressing the aforementioned points.

- We once again would like to thank the reviewer for their time and efforts spent.

Reply to Reviewer #4:

Reviewer #4: The topic is highly relevant to the field of materials science, especially for applications in green energy conversion technologies. The focus on nanoparticle stability and dynamics at the interface of charged perovskite surfaces is timely and of potential interest for researchers working on catalysis, solid state ionics, chemical and electrochemical membranes etc. The paper addresses a significant research gap by elucidating the impact of perovskite defect chemistry on the growth and coalescence behavior of exsolved nanoparticles. The comparative study between acceptor-doped and donor-doped perovskites offers original insights into how defect chemistry influences nanoparticle dynamics, which is less explored in current literature. The paper is well-structured and written with clarity, effectively guiding the reader through the study's rationale, methods, results, and conclusions. The coherent presentation of complex concepts and results facilitates understanding. The figures and plots are of high quality, effectively illustrating key findings and supporting the textual analysis. They are well-integrated into the text, enhancing the paper's overall clarity and impact. Data analysis is meticulous, with a clear correlation between defect chemistry and nanoparticle behavior elucidated through experimental results. The interpretation of data, especially the differences in mass transfer kinetics and the coalescence behavior between STNi and STNNi, is insightful and convincingly argued. Overall, I believe this is a very high quality manuscript and a significant advance in understanding in the field of exsolved materials and nanotechnology for producing robust nanoparticles. I recommend publication, subject to minor points below.

- We sincerely thank the reviewer for their very positive assessment of our manuscript and the clear recommendation for publication. We are happy to provide a more detailed discussion below and in the revised manuscript.

(4.1) *A key aspect that I believe that authors have not made entirely clear is that particle socketing and epitaxy is intimately linked to the use of A-site deficient perovskite (see eg statements on P12). A-site stoichiometric perovskite, as used in the current study, will form other types of metal particles interfaces since B-site exsolution induces the formation of AO phases and/or Ruddlesden Popper phase, thus significantly altering the interface structure and composition and thus stability. Indeed, some of the supplementary data such as Fig S11 indicates that the particle-substrate interface obtained here is atypical of particles exsolved from deficient perovskites. This does not detract from the novelty, but provides a more nuanced discussion around the importance of defect chemistry support for achieving particle stability.*

- We sincerely thank the reviewer for their helpful comment.

Nanoparticle socketing was shown to occur for exsolved nanoparticles in a variety of doped oxides, where the socketing process was previously attributed directly to the growth mechanism of exsolution nanoparticles by cations from the host lattice, allowing for facile interdiffusion between the two phases⁴. However, the reported extent of nanoparticle socketing varies strongly across the literature and despite seminal publications on the topic^{4,8}, a universal mechanistic explanation on the origin of socket formation and the controllability of the socket depth and related properties is lacking.

We agree that the extent of interdiffusion between the two phases may be enhanced for A-deficient perovskites often used as efficient parent oxides for the synthesis of exsolution catalysts in comparison to the stoichiometric oxides investigated in the present study. However, there are different reports in the literature indicating that the situation is more complex.

For example, Ni nanoparticles that were prepared by solution deposition on an A-site deficient oxide support did not reveal indications for socketing⁴. In contrast, however, the formation of nanoparticle sockets was shown for Ni nanoparticles prepared on stoichiometric BaZr_{0.9}Y_{0.1}O₃-based oxides irrespective of the (exsolution and deposition) preparation method⁵ and has been even demonstrated for A-site excess LaFeO₃ thin films with Pd nanoparticles being prepared by dewetting experiments⁶. Furthermore, socketing was observed exsolved nanoparticles in stoichiometric SrIr_{0.005}Ti_{0.995}O₃ perovskites^{7,8}.

We have revised our discussion to provide a more detailed discussion on the matter. Our statement now reads as follows (line 456 – line 469).

“It is worth noting that nanoparticle socketing was shown to occur for exsolved nanoparticles in a variety of doped oxides, where the socketing process was previously attributed directly to the growth mechanism of exsolution nanoparticles by cations from the host lattice, allowing for facile interdiffusion between the two phases⁴. However, the reported extent of nanoparticle socketing varies strongly across the literature and despite seminal publications on the topic^{4,8}, a universal mechanistic explanation on the origin of socket formation and the controllability of the socket depth and related properties is lacking. Oftentimes, A-cation deficient perovskites are used as efficient parent oxides for the synthesis of exsolution catalysts potentially enhancing the extent of interdiffusion in comparison to the stoichiometric oxides investigated in the present study. The formation of nanoparticle sockets, however, was shown to occur for Ni nanoparticles prepared on stoichiometric $\text{BaZr}_{0.9}\text{Y}_{0.1}\text{O}_3$ -based oxides irrespective of the (exsolution and deposition) preparation method⁵, for exsolved nanoparticles in stoichiometric $\text{SrIr}_{0.005}\text{Ti}_{0.995}\text{O}_3$ perovskites^{7,8}, and has been even demonstrated for A-site excess LaFeO_3 thin films with Pd nanoparticles being prepared by dewetting experiments⁶.”

References [8] and [6] have been included in the revised manuscript as references [58] and [59].

(4.2) *Additionally, while the paper effectively correlates defect chemistry with nanoparticle dynamics, it could benefit from a deeper mechanistic insight. Specifically, the discussion around charge carrier concentration and its impact on surface energy modifications could be expanded. Including theoretical models or simulations that support the experimental observations could enrich the discussion.*

- We thank the reviewer for their suggestion. In the current study, we start from established defect chemical understanding of complex oxides and particularly SrTiO_3 -based oxides, in which acceptors (such as incorporated Ni in STNi) are predominantly compensated by oxygen vacancies. Therefore, the concentration of oxygen vacancies will be orders of magnitude larger than in the co-doped case, where the effective doping is donor-type (as indicated by the negligible oxygen diffusion, Fig. 1). Combined with the existing knowledge on accumulation of oxygen vacancies in the surface layer of SrTiO_3 (due to space charge formation^{14,15}), it is clear that a larger concentration of oxygen vacancies yields a significantly reduced number of possible Ni-O bonds for the exsolved nanoparticles and, thus, a lower thermal stability. We agree that more detailed simulations could further help to understand the microscopic mechanism of particle motion, growth, and coalescence. Such dynamic simulations, however, are not straightforward as they will require an in-depth investigation of the particle shape and its deformation during motion and growth. An experimental approach to address these atomistic parameters *in-situ* reflects work in progress but goes beyond the scope of the current manuscript.

References

1. Weber, M. L. *et al.* Space charge governs the kinetics of metal exsolution. *Nature Mater* **23**, 406–413; 10.1038/s41563-023-01743-6 (2024).
2. Syed, K., Wang, J., Yildiz, B. & Bowman, W. J. Bulk and surface exsolution produces a variety of Fe-rich and Fe-depleted ellipsoidal nanostructures in $\text{La}_{0.6}\text{Sr}_{0.4}\text{FeO}_3$ thin films. *Nanoscale* **14**, 663–674; 10.1039/d1nr06121f (2022).
3. Wang, J. *et al.* Exsolution Synthesis of Nanocomposite Perovskites with Tunable Electrical and Magnetic Properties. *Adv. Funct. Mater.* **32**, 2108005; 10.1002/adfm.202108005 (2022).
4. Neagu, D. *et al.* Nano-socketed nickel particles with enhanced coking resistance grown in situ by redox exsolution. *Nature communications* **6**, 8120; 10.1038/ncomms9120 (2015).
5. Jennings, D., Ricote, S., Santiso, J., Caicedo, J. & Reimanis, I. Effects of exsolution on the stability and morphology of Ni nanoparticles on BZY thin films. *Acta Materialia* **228**, 117752; 10.1016/j.actamat.2022.117752 (2022).
6. Katz, M. B. *et al.* Self-regeneration of Pd-LaFeO₃ catalysts: new insight from atomic-resolution electron microscopy. *Journal of the American Chemical Society* **133**, 18090–18093; 10.1021/ja2082284 (2011).
7. Cali, E. *et al.* Exsolution of Catalytically Active Iridium Nanoparticles from Strontium Titanate. *ACS applied materials & interfaces* **12**, 37444–37453; 10.1021/acsami.0c08928 (2020).
8. Cali, E. *et al.* Real-time insight into the multistage mechanism of nanoparticle exsolution from a perovskite host surface. *Nature communications* **14**, 1754; 10.1038/s41467-023-37212-6 (2023).
9. Santaya, M. *et al.* Exsolution versus particle segregation on (Ni,Co)-doped and undoped $\text{SrTi}_{0.3}\text{Fe}_{0.7}\text{O}_{3-\delta}$ perovskites: Differences and influence of the reduction path on the final system nanostructure. *International Journal of Hydrogen Energy* **48**, 38842–38853; 10.1016/j.ijhydene.2023.06.203 (2023).
10. Winterbottom, W. Equilibrium shape of a small particle in contact with a foreign substrate. *Acta Metallurgica* **15**, 303–310; 10.1016/0001-6160(67)90206-4 (1967).
11. Waser, R. Bulk Conductivity and Defect Chemistry of Acceptor-Doped Strontium Titanate in the Quenched State. *J American Ceramic Society* **74**, 1934–1940; 10.1111/j.1151-2916.1991.tb07812.x (1991).
12. Kyriakou, V. *et al.* Plasma Driven Exsolution for Nanoscale Functionalization of Perovskite Oxides. *Small methods* **5**, e2100868; 10.1002/smt.202100868 (2021).
13. Shin, E. *et al.* Ultrafast Ambient-Air Exsolution on Metal Oxide via Momentary Photothermal Effect. *ACS nano* **16**, 18133–18142; 10.1021/acsnano.2c05128 (2022).
14. Souza, R. A. de. The formation of equilibrium space-charge zones at grain boundaries in the perovskite oxide SrTiO_3 . *Physical chemistry chemical physics : PCCP* **11**, 9939–9969; 10.1039/B904100A (2009).
15. Souza, R. A. de, Metlenko, V., Park, D. & Weirich, T. E. Behavior of oxygen vacancies in single-crystal SrTiO_3 : Equilibrium distribution and diffusion kinetics. *Phys. Rev. B* **85**; 10.1103/PhysRevB.85.174109 (2012).

Point-by-point response letter regarding revision of manuscript NCOMMS-23-62974A

We sincerely thank the referees for their time and expertise contributed reviewing our manuscript. We are glad that all concerns raised by reviewers #1, #2 and 4# are resolved and for their recommendation to publish our work in *Nature Communications*. Below, we give a point-by-point response to all remarks of reviewer #3, in hope to reach a consensus based on a second substantial revision of our manuscript.

Reply to Reviewer #3:

Reviewer #3: The authors addressed in their revised version most of the points of my assessment, but there is a major topic which has not yet been resolved. The authors use the experimental findings presented in the main text of this work (and not the findings of the supporting information) to make generalized conclusions and strong statements about the limited applicability of exsolution synthesis route for electrodes that require high oxygen mobility. However, this generalized conclusion lacks foundation and is not appropriately justified. To be more specific, my concern relates to the following phrases which appear in the abstract and conclusion sections. Abstract: Our analysis indicates that the low thermal stability of exsolved nanoparticles at the acceptor-doped perovskite surface is associated to a large oxygen vacancy concentration at the nanoparticle-oxide interface, hampering the applicability of the exsolution synthesis route for catalysts that require a fast oxygen exchange kinetics. Conclusions: The results of the present study may indicate that the application of exsolution catalysts may be limited to specific material combinations, where the fabrication of metal nanoparticles at the surface of fast oxygen ion conductors as a material class that is interesting for applications in solid oxide cells, appears to be highly challenging. The need for a low surface oxygen vacancy concentration to stabilize exsolved nanoparticles hence requires strongly donor-type perovskites with low oxygen ion conductivity and typically high electron conductivity or modifications in the respective fuel composition applied in the respective device. I strongly believe that the message that the authors are trying to convey is inappropriate and misleading, for the following reasons. First of all, the present study does demonstrate the low stability of the exsolved particles on perovskites with high oxygen mobility, but only when unrealistic conditions are applied i.e. dry hydrogen environment. The dry hydrogen conditions are unrealistic since the exsolution concept with high oxygen mobility perovskites is mainly applicable for oxygen conducting solid oxide cells (SOCs). However, the fuel electrode in SOCs typically involves wet conditions, for instance: (i) in the fuel cell mode of SOCs, water is formed in the fuel electrode during operation, (ii) in the water electrolysis mode of SOCs, hydrogen and water co-exist, (iii) in the carbon dioxide electroreduction mode of SOCs, water is in the feed. In oxygen electrodes, exsolved nanoparticle decorated electrodes operate in the absence of hydrogen and thus the findings of this work are also not applicable. Moreover, the results of the present study in wet hydrogen conditions (presented in the supplementary information), which are the realistic scenario, showed that exsolved nanoparticles are stable in both evaluated classes of materials (i.e. with high and low oxygen mobility). Thus, the results of the present study do not support the generalized statements made in the abstract and conclusions and thus I cannot recommend the publication of this work. Based on these points, I would invite the authors to make a second revision if they think that they can address these concerns and if not my recommendation would be the rejection of this work.

- We thank the reviewer for their valuable comment, and we agree that is necessary to clearly communicate that the operation conditions of SOCs may have a considerable impact on the coalescence behaviour of exsolved nanoparticles due to an increased humidity of the reaction atmosphere as we have demonstrated in our original manuscript (cf. discussion in line 512 – 524).

We would like to emphasize that we have specifically addressed this matter in our original manuscript (line 539 – line 542). For the convenience of the referee, we repeat our statement in the following.

“The highly reducing environment of the cathodic reaction in electrolysis would pose a particular challenge, as the nanoparticle stability may be adversely affected by the cathodic overpotential and local gas conversion effects. Whether such effects are mitigated by the high humidity present under operation conditions remains to be investigated.”

However, we fully acknowledge that generalized statements need to be avoided in the abstract and in the conclusion of the manuscript. Accordingly, we have revised both sections to provide a more nuanced discussion.

Accordingly, our abstract now reads as follows.

“[...] Our analysis indicates that the low thermal stability of exsolved nanoparticles at the acceptor-doped perovskite surface is associated with a high oxygen vacancy concentration at the nanoparticle-oxide interface. For catalysts that require fast oxygen exchange kinetics, exsolution synthesis routes in dry hydrogen conditions may hence lead to rapid degradation, while humid reaction conditions may mitigate this failure mechanism.

Moreover, our conclusion now reads as follows.

“[...] The results of the present study may indicate that the application of exsolution catalysts under strongly reducing atmosphere (such as dry hydrogen) may be limited to specific material combinations. Particularly, the fabrication of metal nanoparticles at the surface of fast oxygen ion conductors as a material class that is interesting for applications in solid oxide cells, appears to be highly challenging. The need for a low surface oxygen vacancy concentration to stabilize exsolved nanoparticles hence requires strongly donor-type perovskites with low oxygen ion conductivity and typically high electron conductivity or a humid gas composition present in the respective device. The wet hydrogen atmosphere that is often present under SOC operation, hence might result in a suppressed coalescence dynamics as a result of an increased oxygen partial pressure.”